# Accurate Prediction of Protein Structural Flexibility by Deep Learning Integrating Intricate Atomic Structures and Cryo-EM Density Information

Xintao Song[1,2,3,7], Lei Bao [4,7], Chenjie Feng[5], Qiang Huang[1], Fa Zhang [6]✉, Xin Gao [3]✉ & Renmin Han [1,2]✉

The dynamics of proteins are crucial for understanding their mechanisms. However, computationally predicting protein dynamic information has proven challenging. Here, we propose a neural network model, RMSF-net, which outperforms previous methods and produces the best results in a large-scale protein dynamics dataset; this model can accurately infer the dynamic information of a protein in only a few seconds. By learning effectively from experimental protein structure data and cryo-electron microscopy (cryo-EM) data integration, our approach is able to accurately identify the interactive bidirectional constraints and supervision between cryo-EM maps and PDB models in maximizing the dynamic prediction efficacy. Rigorous 5-fold cross-validation on the dataset demonstrates that RMSF-net achieves test correlation coefficients of $0.746 \pm 0.127$ at the voxel level and $0.765 \pm 0.109$ at the residue level, showcasing its ability to deliver dynamic predictions closely approximating molecular dynamics simulations. Additionally, it offers real-time dynamic inference with minimal storage overhead on the order of megabytes. RMSF-net is a freely accessible tool and is anticipated to play an essential role in the study of protein dynamics.

The dynamics of proteins play a crucial role in understanding their mechanisms[1-3]. Currently, most proteins are solved by cryo-electron microscopy (cryo-EM) technology, where the macromolecular structures are represented by 3D density maps[4-6].

Due to the low resolution and signal-to-noise ratio of the original two-dimensional particle images in cryo-EM analysis, small conformation changes cannot be resolved during reconstruction[4]. Although subspace clustering methods based on implicit features have been proposed to separate very different conformations[7-9], cryo-EM maps reconstructed from a series of two-dimensional images categorized within the same group still contain dynamic information concerning structural fluctuations and conformation changes within macromolecules.

Deep learning methods have been widely applied to the automatic analysis of cryo-EM maps, such as substructure segmentation[10-12], de novo atomic model construction[13-15] and prediction-assisted

[1]Research Center for Mathematics and Interdisciplinary Sciences (Ministry of Education Frontiers Science Center for Nonlinear Expectations), Shandong University, Qingdao, China. [2]BioMap Research, Menlo Park, CA, USA. [3]King Abdullah University of Science and Technology (KAUST), Computational Bioscience Research Center (CBRC), Computer, Electrical and Mathematical Sciences and Engineering (CEMSE) Division, Thuwal, Saudi Arabia. [4]School of Public Health, Hubei University of Medicine, Shiyan, China. [5]College of Medical Information and Engineering, Ningxia Medical University, Yinchuan, China. [6]School of Medical Technology, Beijing Institute of Technology, Beijing, China. [7]These authors contributed equally: Xintao Song, Lei Bao. ✉e-mail: zhangfa@bit.edu.cn; xin.gao@kaust.edu.sa; hanrenmin@sdu.edu.cn

modeling[16,17]. Currently, given a high-resolution cryo-EM map, it is not a difficult task to build a PDB model exactly from the cryo-EM map[18]. However, these built PDB models do not consider the dynamic information, and calculating dynamic information from the PDB model remains a difficult molecular dynamics (MD) simulation task, which usually requires considerable computational resources and time[19–21].

The root-mean-square fluctuation (RMSF) measures the average deviation of a protein residue over time from a reference position, which is a very important dynamic index that reflects the portions of a macromolecular structure that fluctuate from their mean structure the most (or least). Recently, a deep learning method called DEFMap was proposed to directly extract dynamic information from cryo-EM maps[22]. However, DEFMap employs only 34 proteins as the training and testing dataset, which may make the model overfit the training set and unusable in practical applications. In addition, the cryo-EM maps contain both static structure information of the protein and local density changes due to structural fluctuations, i.e., inhomogeneity of the density map due to flexibility. In DEFMap, the authors calculated the correlation coefficients of local density and local resolution with RMSF. Both exceed 0.4 in more than half of the data. This indicates a correlation between the local density distribution of cryo-EM maps and structural flexibility. However, due to the black-box nature of neural networks, whether there is indeed a pattern between local density distribution and flexibility in the network and what role the static structure information plays in it is still unclear. Furthermore, whether we can design a more powerful model while studying these factors remains to be explored.

In this work, we present RMSF-net, a neural network model for cryo-EM density maps that can accurately infer the dynamic information of a protein within only a few seconds, by fully utilizing the cryo-EM density and PDB model information. Along with the cryo-EM maps, RMSF-net utilizes the PDB model as an additional input to produce RMSF predictions that are very close to the MD simulation results. In addition, a large-scale protein dynamics dataset is built for the training and validation of RMSF-net, in which 335 cryo-EM structure entries with fitted PDB models are selected and the corresponding MD simulations are performed. Comprehensive experimental results demonstrate the efficiency and effectiveness of RMSF-net. In particular, RMSF-net shows superior performance on the test set through a rigorous 5-fold cross-validation, achieving a correlation coefficient of $0.746 \pm 0.127$ to the MD simulation results, representing a 15% improvement over DEFMap and a 10% improvement over the baseline. Specifically, RMSF-net is built into a freely available open-source tool for the structural biology society which can be easily accessed as a Python package.

## Results

### Overview of RMSF-net procedure

We propose a deep learning approach named RMSF-net to analyze protein dynamics based on cryo-electron microscopy (cryo-EM) maps. The primary objective of this method is to predict the RMSF of local structures (residue, atoms) within proteins. RMSF is a widely used measure to assess the flexibility of molecular structures in MD analysis and defined by the following equation:

$$RMSF = \sqrt{\frac{1}{T}\sum_{t=1}^{T}\left(x(t)-\tilde{x}\right)^2}$$

where $x$ represents the real-time position of atoms or residues, $t$ represents time and $\tilde{x}$ represents the mean position over a period of time $T$. In addition to the experimental cryo-EM maps, RMSF-net incorporates fitted PDB models, which represent the mean structures of fluctuating proteins. A schematic overview of RMSF-net is depicted in Fig. 1a. The cryo-EM map and PDB model are initially combined to create a dual feature pair. The PDB models are converted into voxelized density maps using the "MOLMAP" tool in UCSF Chimera[23] to facilitate seamless integration with cryo-EM maps. Subsequently, both the density grids of the cryo-EM maps and the PDB simulated maps are divided into uniform-sized density boxes ($40 \times 40 \times 40$) with a stride of 10. The corresponding density boxes from the mapping pair are concatenated to form a two-channel feature input for the neural

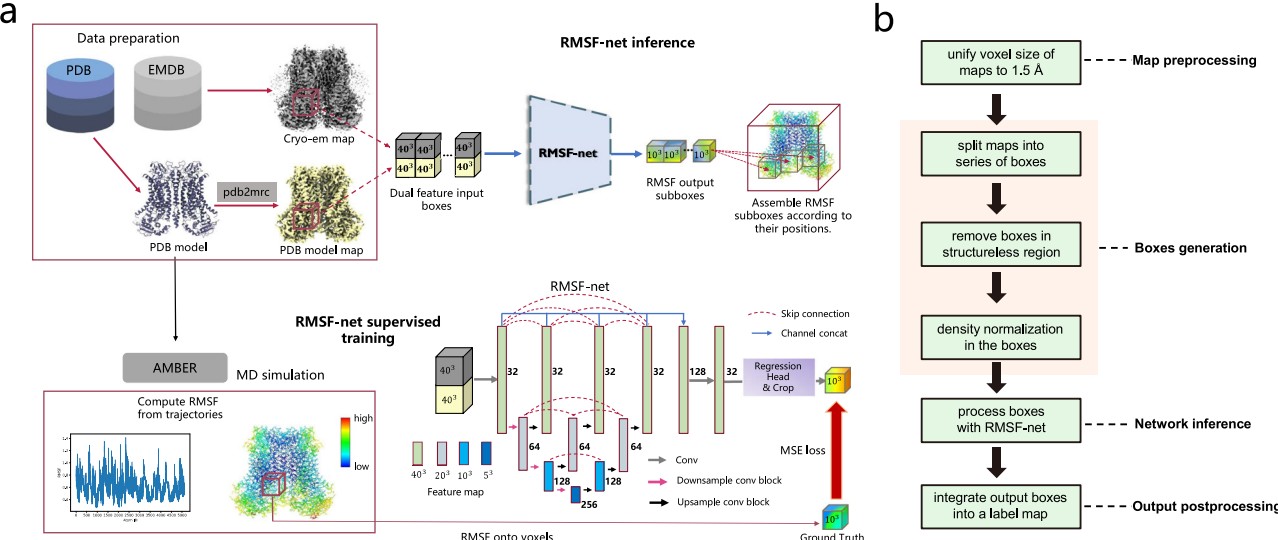

**Fig. 1 | RMSF-net implementation. a** Overview of RMSF-net. The data preparation and RMSF inference for RMSF-net are illustrated in the upper section. Cryo-EM maps and their fitted atomic structure models were obtained from the EMDB and PDB databases. The PDB models were simulated as density maps resembling cryo-EM maps. Both the cryo-EM map and PDB simulated map were then segmented into 40-cubic density boxes. The density boxes with matching positions from the pair were concatenated into a two-channel tensor and input to the 3D CNN of RMSF-net to infer the RMSF for atoms in the central $10^3$ voxels (subboxes). The RMSF prediction across the entire map was obtained by combining predictions from these subboxes. The lower section depicts the RMSF-net supervised training process. The RMSF-net neural network architecture is shown in the lower right, with the number of channels indicated alongside the hidden feature maps. With Unet + + (L3) as the backbone, a regression head and crop operation were added. The ground truth RMSF for training RMSF-net was derived from MD simulations, as illustrated in the lower left. **b** Data processing of maps in RMSF-net.

network (RMSF-net) to predict the RMSF of atoms within the central subbox ($10 \times 10 \times 10$). RMSF-net is a three-dimensional convolutional neural network comprising two interconnected modules. The primary module employs a Unet + +(L3) architecture[24] for feature encoding and decoding on the input density boxes. The other module utilizes 1-kernel convolutions for regression on the channels of the feature map generated by the Unet + + backbone. A center crop is then applied to the regression module output to obtain the central RMSF subboxes, where the voxel values correspond to the RMSF of the atoms contained within them. Finally, the RMSF sub-boxes are spatially merged into an RMSF map using a merging algorithm.

RMSF-net incorporates several data processing strategies for maps, as illustrated in Fig. 1b. First, to ensure a consistent spatial scale, all the cryo-EM maps were resampled using the "ndimage.zoom" module from SciPy[25] to obtain a uniform voxel size of 1.5 Å, which is approximately the size of a Cα atom. Second, a screening algorithm is applied to retain only those boxes that encompass atoms within the central subbox to avoid processing unnecessary boxes located in structure-free regions. This strategy significantly improves the efficiency of RMSF-net when dealing with large cryo-EM maps. The retained box indices are recorded for the subsequent merging algorithm. In addition, the voxel densities within the boxes are normalized to a range of [0, 1] before being input into the network, thus mitigating density distribution variations across cryo-EM maps. Voxel density normalization is achieved through the following process: within each box, any density values less than 0 are set to 0, and then divided by the maximum density value within the box, thus scaling the voxel density to a range from 0 to 1.

## Dataset construction
We created a high-quality protein dataset with 335 entries for training and evaluating RMSF-net. The dataset was constructed by selecting data from EMDB[26] and PDB[27]. As of November 2022, EMDB contained over 23,593 deposited entries, with more than half being high-resolution maps with resolutions ranging from 2–4 Å. We focused on maps within this range. Initially, we included a high-resolution cryo-EM map dataset from EMNUSS[12] which underwent rigorous screening in EMDB and PDB prior to October 2020 and consisted of 468 entries. In addition, we performed data selection on deposited cryo-EM maps and PDB models from October 2020 to November 2022 to incorporate newly deposited data. The selected data had to meet specific criteria, including a well-fitting cryo-EM map and PDB model with a fitness above 0.7 (measured by the correlation between the PDB simulated map and cryo-EM map); the proteins had to contain at least one alpha helix or beta-strand, with no missing chains or nucleic acids. We further filtered these data by applying a clustering procedure to remove redundancy. Using the K-Medoids[28] algorithm with a k value of 50, we defined the distance between two proteins as the maximum distance between any two chains from each protein, where chain distances were determined by sequence identity. After clustering, we selected the 50 medoids and added them to the dataset. Finally, out of the remaining 518 entries, 335 were successfully subjected to MD simulations, resulting in the RMSF-net dataset.

## Molecular dynamics simulation
RMSF-net employs a supervised training approach, requiring labeled RMSF values derived from MD simulations[19]. We conducted MD simulations on the PDB models of the dataset following a standardized procedure using Assisted Model Building with Energy Refinement (AMBER)[20], which consists of four stages: energy minimization, heating, equilibration, and production runs. To focus on local structure fluctuations around specific protein conformations, we configured the production run for 30 nanoseconds. Specifically, the initial atomic coordinates of the proteins were set to the original PDB model coordinates. Small molecule ligands in all complexes were removed to

purely study the characteristics of proteins. Each system was immersed in a truncated octahedron box filled with TIP3P[29] water molecules (at least a 12 Å buffer distance between the solute and edge of the periodic box). Based on the charge carried by the protein, $Na^+$ or $Cl^-$ ions were placed randomly in the simulation box to keep each system neutral. An additional 150 mM NaCl solution was added to all systems according to the screening layer tally by the container average potential method[30] to match the experimental conditions better. All MD simulations were performed using the AMBER 20 software package[20,31] on NVIDIA Tesla A100 graphics cards. The parameters for $Na^+$ and $Cl^-$ ions were derived from the previous work by Joung et al.[32]. The parameters used for the protein structure were AMBER ff14SB force field[33]. Each system was energy minimized using the conjugate gradient method for 6000 steps. Then, the systems were heated using the Langevin thermostat[34] from 0 to 300 K in 400 ps using position restraints with a force constant of 1000 kcal · $mol^{-1}$ · $Å^{-2}$ to the protein structure (NVT ensemble, $T = 300$ K). Subsequently, each system was gradually released in 5 ns (spending 1 ns each with position restraints of 1000, 100, 10, 1, and 0 kcal · $mol^{-1}$ · $Å^{-2}$) using the NPT ensemble ($P = 1$ bar, $T = 300$ K) before a production run. Afterward, the final structure of each system was subjected to a 30 ns MD simulation at constant temperature (300 K) and pressure (1 bar) with periodic boundary conditions and the particle mesh Ewald (PME) method[35] We used the isotropic Berendsen barostat[36] with a time constant of 2 ps to control constant pressure. The protein structure was completely free in the solutions during the equilibration and production process. Simulations were run with an integration step of 2 fs, and bond lengths for hydrogen atoms were fixed using the SHAKE algorithm[37]. PME electrostatics were calculated with an Ewald radius of 10 Å, and the cutoff distance was also set to 10 Å for the van der Waals potential.

## Model training and validation
After simulation, the trajectories were processed and analyzed using the built-in Cpptraj module of AMBER Tools package[38]. We first removed the translational and rotational motion of all protein molecules to ensure a rigorous comparison between different trajectories. Then, the average structure of each protein (only heavy atoms) was calculated as a reference structure. Afterward, each conformation in the trajectory was aligned to the reference structure and RMSF of the protein molecule was output. These computed RMSF values were subsequently mapped onto voxels of cryo-EM maps to serve as the ground truth for training and evaluating RMSF-net.

For the training of RMSF-net, we utilized a masked mean squared error (MSE) loss function to compute the loss between the predicted RMSF and ground truth RMSF on labeled voxels of the output sub-boxes. The training spanned 100 epochs with a batch size of 32, and we employed the Adam optimizer[39] with a learning rate of 0.004. Several techniques were implemented to mitigate overfitting, including Kaiming weight initialization[40], learning rate decay, and early stopping. If the validation loss did not decrease for 10 consecutive epochs, the learning rate was halved. If it did not decrease for 30 epochs, training was terminated, and the model with the minimum validation loss was saved. We applied rotation and mirroring augmentation to the training set to account for the lack of rotational and mirror symmetry in convolutional networks, increasing the training data eightfold. The training of RMSF-net was conducted on two NVIDIA Tesla A100 graphics cards, typically lasting 5–8 h. Following training, we conducted RMSF predictions on the test set to evaluate the performance of RMSF-net.

We employed a five-fold cross-validation approach to assess the performance of this method. The dataset was randomly divided into five equal partitions, with one partition used as the test set each time, and the remaining four partitions served as the training and validation sets. In particular, the division was based on the maps rather than the segmented boxes in order to ensure independence between these sets. The training and testing process was repeated five times, and every

data entry was tested once to obtain the method's performance on the entire dataset. To prevent overfitting during model training, the training and validation sets were set at a ratio of 3:1. During testing, the correlation coefficients between the predicted RMSF and the ground truth (RMSF values derived from MD simulations) were computed as the evaluation metric. The correlation coefficients were computed at two levels: voxel level, corresponding to RMSF on the map voxels, and residue level, corresponding to RMSF on the PDB model residues (obtained by averaging RMSF on the corresponding atoms). We defaulted to using the correlation coefficient at the voxel level when analyzing and comparing model performance unless otherwise specified. In addition, we employed the correlation coefficient at the residue level when discussing the protein test cases.

### Baselines

As a baseline, we initially used only the cryo-EM map intensity as the single-channel input to the neural network, referred to as RMSF-net_cryo. Cross-validation using RMSF-net_cryo on the dataset revealed an average correlation coefficient of 0.649 and a bias of 0.156. We also performed cross-validation using the prior method DEFMap method for comparison. DEFMap reported a test correlation of approximately 0.7 on its dataset. However, its dataset includes only 34 proteins and the dataset used in our study is more diverse and significantly larger. Therefore, we applied the DEFMap pipeline to our dataset to ensure fair comparisons. Notably, DEFMap employed different data preprocessing strategies and neural networks. During its preprocessing, a low-pass filter was adopted to standardize the resolution of the cryo-EM maps. In addition, the neural network it used took 10-cubic subvoxels as input and outputted the RMSF of the central voxel. We strictly followed DEFMap's procedures and network for training and testing. The results showed an average correlation coefficient of 0.6 and a bias of 0.171. Through comparison, it is evident that RMSF-net_cryo exhibits superior performance compared to DEFMap.

### Enhancing interpretability of dynamics inferred from cryo-EM intensity

Although RMSF-net_cryo performed better than DEFMap with our designed network and data processing strategies, it still relies on neural networks to directly establish patterns between cryo-EM maps and flexibility. What role the structural information plays in this process remains unknown. This prompted us to divide dynamic prediction via cryo-EM maps into two sequential steps: first, structural information extraction, and second, dynamic prediction based on the extracted structural information.

To accomplish the extraction of structural information, as depicted in Fig. 2a, we introduced an Occ-net module. This module predicts the probabilities of structural occupancy on cryo-EM map voxels using a 3D convolutional network. Both input and output dimensions were set to $40^3$. For training and evaluating Occ-net, we utilized PDB models to generate structure annotation maps as the ground truth, where voxels were categorized into two classes: occupied by structure and unoccupied by structure. The details of this network and data annotation process are provided in the Supplementary Information (section "Structure of Occ-net and the data annotation process"). Cross-entropy loss function was employed during training, with class weights set to 0.05:0.95 to address class imbalance. Once this training stage was completed, Occ-net parameters were fixed, and the second stage of training commenced. In the second stage, the two-channel classification probabilities output by Occ-net were input into the dynamics extraction module to predict the RMSF for the central $10^3$ voxels, which is consistent with the RMSF-net approach.

This model is referred to as Occ2RMSF-net, and cross-validation was conducted on it. After training, we first assessed the performance of Occ-net by calculating the precision, recall, and F1-score at the voxel level for the positive class (structure class) on the test set. This evaluation involves six classification thresholds, ranging from 0.3 to 0.8. As depicted in Fig. 2b, achieving high precision and recall simultaneously was challenging due to severe class imbalance and noise. A relative balance was achieved at the threshold of 0.7, where the F1 score reached its highest value of 0.581. Regarding the final output RMSF, the correlation between the Occ2RMSF-net predictions and the ground truth on the dataset is $0.662 \pm 0.158$, showing a slight improvement compared to RMSF-net cryo. Figure 2c displays the scatter plot of the test correlation of data for Occ2RMSF-net and RMSF-net cryo. The two models exhibited similar performance on most of the data points, with Occ2RMSF-net slightly outperforming RMSF-net_cryo overall. This highlights the critical role of structure information from cryo-EM maps for predicting the RMSF in the network and enhances the interpretability of methods like DEFMap and RMSF-net_cryo.

### Performance of RMSF-net that incorporates PDB information

Inspired by the above results, we incorporated PDB models representing precise structural information and integrated them into our method in a density map-like manner, i.e., simulated density maps generated based on PDB models. We employed two approaches to input the PDB simulated maps into the network. First, the PDB simulated map was taken as a single-channel feature input into the neural network, referred to as RMSF-net_pdb. Second, the PDB simulated map was transformed into a binary encoding map representing occupancy of the structure to highlight tertiary structural features: a threshold of 3 σ(σ represents r.m.s.d of the PDB simulated map density) was chosen, where voxels with densities above the threshold were encoded as 1 and the others as 0. This encoding map was then converted into a two-channel one-hot input to the network, known as RMSF-net_pdb01. The same cross-validation was applied to these two models. Results showed that RMSF-net_pdb achieved a test correlation coefficient of $0.723 \pm 0.117$, and RMSF-net_pdb01 achieved $0.712 \pm 0.112$. These two approaches demonstrated significantly better performance than the above cryo-EM map-based methods, further demonstrating the strong correlation between protein structure topology and flexibility.

We further combined the information from the PDB structure and cryo-EM map and input them into the network, which is the main method proposed in this work. We refer to this method, along with the neural network it employs, simply as RMSF-net. As outlined in the "Overview of RMSF-net procedure" section, RMSF-net takes the dual-channel feature of density from the cryo-EM map and PDB simulated map at the same spatial position as input, while the main part of the network remains the same as RMSF-net_cryo and RMSF-net_pdb. Conducting the same cross-validation on RMSF-net, the results revealed an average correlation coefficient of 0.746, with a median of 0.767 and a standard deviation of 0.127.

RMSF-net demonstrated an approximate 10% improvement compared to the baseline of RMSF-net_cryo, and a 15% enhancement over DEFMap. Figure 2d presents a comparison of the distribution of data quantities at different test correlation levels for the three methods. Overall, the two cryo-EM map-based methods (DEFMap and RMSF-net_cryo) exhibit similar distribution shapes, while the distribution of RMSF-net is more concentrated, focusing on the range between 0.6 and 0.9. Nearly half of the data points cluster around 0.7 to 0.8, and close to one-third fall between 0.8 and 0.9. In comparison, the two PDB-based methods in Fig. 2f exhibit similar distributions with RMSF-net. This suggests that the structure information from PDB models plays a primary role in the ability of RMSF-net to predict flexibility. On the other hand, RMSF-net further outforms the PDB-based methods through combination with information from cryo-EM maps, indicating image features related to structural flexibility in the cryo-EM

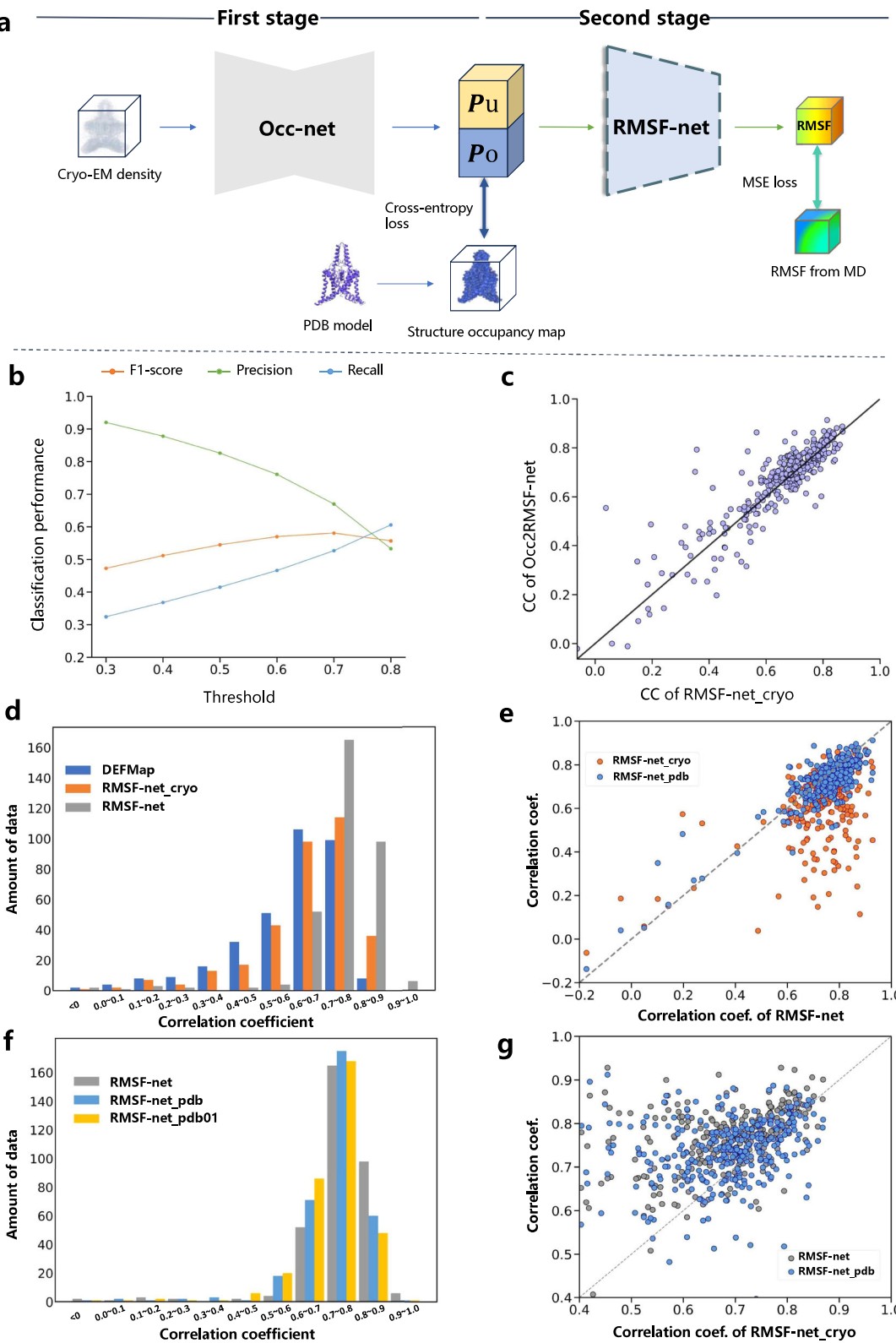

**Fig. 2 | Comparison of performance between RMSF-net and other relevant approaches. a** Overview of Occ2RMSF-net. In the first stage, the cryo-EM density ($40^3$) is input into Occ-net to predict the probabilities ($40^3$) of structure occupancy on the voxels, with $P_u$ denoting the probabilities of voxels being occupied by the protein structure and $P_o$ denoting that of not being occupied by the structure. Then in the second stage, the two-channel probabilities are input into RMSF-net to predict the RMSF on the center $10^3$ voxels. **b** Test performance of Occ2net. For six classification thresholds from 0.3 to 0.8, the precisions, recalls and F1-scores of the positive class (structure occupied) on the test set were computed and are shown in the plot. **c** Comparison of RMSF prediction performance between Occ2RMSF-net and RMSF-net_cryo on the dataset. CC is an abbreviation for correlation coefficient. **d** Count distribution of test correlation coefficients for DEFMap, RMSF-net_cryo, and RMSF-net on the dataset. **e** Data distribution of correlation coefficients for RMSF-net_cryo and RMSF-net_pdb relative to RMSF-net on the dataset. **f** Count distribution of test correlation coefficients for RMSF-net_pdb, RMSF-net_pdb01, and RMSF-net on the dataset. **g** Data distribution of correlation coefficients for RMSF-net and RMSF-net_pdb relative to RMSF-net_cryo on data points where the test correlation coefficients with RMSF-net_cryo are above 0.4. The color for each method in **d**, **e**, **f** and **g** is shown in the legend.

**Table 1 | Performance of different RMSF prediction methods on the dataset**

| Corre. Coff. | Statistics | DEFMap | RMSF-net_cryo | RMSF-net | RMSF-net_pdb | RMSF-net_pdb01 |
|---|---|---|---|---|---|---|
| Voxel level | avg | 0.6 | 0.649 | **0.746** | 0.723 | 0.712 |
| | med | 0.648 | 0.688 | **0.767** | 0.74 | 0.726 |
| | std | 0.171 | 0.156 | 0.127 | 0.117 | **0.112** |
| Residue level | avg | 0.622 | 0.671 | **0.765** | 0.742 | 0.732 |
| | med | 0.67 | 0.708 | **0.78** | 0.758 | 0.746 |
| | std | 0.159 | 0.148 | 0.109 | 0.101 | **0.097** |

Bold indicates the best of the statistical value among these models.

map make an effective auxiliary. Regarding the robustness of these approaches, Table 1 demonstrates that RMSF-net_pdb and RMSF-net_pdb01 exhibited less deviation on the test set compared to RMSF-net, while RMSF-net_cryo displayed the highest deviation. This indicates the flexibility-related information in cryo-EM maps is unstable compared to that in PDB models, which might be caused by noise and alignment errors in cryo-EM maps.

**Anomalies**

The experimental results above prove that the combination of cryo-EM map and PDB model results in the superior performance of RMSF-net. As shown in Fig. 2e, the prediction of RMSF-net is better in most cases, comparing models utilizing only the cryo-EM map or only the PDB model. Because the PDB models are built from the corresponding cryo-EM maps, their spatial coordinates are naturally aligned, and their structural information is consistent. Moreover, the PDB model built from the cryo-EM map corresponds precisely to the average position of the structure, and the cryo-EM map reconstructed from multiple particles in the sample corresponds to the information of multiple instantaneous conformations. By combining the 'expectation' and conformational 'variance' from the two sources, we believe that this structural consistency and complementarity create an alignment effect, and promote the superior performance of RMSF-net. However, structural deviations may exist between the PDB model and the cryo-EM map in some instances, or the PDB model may only partially occupy the cryo-EM map. These anomalies might lead to subpar performance of RMSF-net_cryo compared to RMSF-net and RMSF-net_pdb. To exclude the influence of these factors, we performed dataset filtering by excluding data points with test correlations below 0.4 for RMSF-net_cryo and compared the three models on the filtered dataset. Figure 2g shows that RMSF-net and RMSF-net_pdb still demonstrated better performance overall compared to RMSF-net_cryo on the filtered dataset. The test correlations for the three models were $0.760 \pm 0.084$, $0.733 \pm 0.083$, and $0.684 \pm 0.1$, respectively. When the filtering threshold was increased to 0.5, the correlations for the three models were $0.761 \pm 0.08$, $0.734 \pm 0.08$, and $0.698 \pm 0.084$, respectively, showing consistent results.

**Examples and case study**

Figure 3a showcases RMSF-net predictions for three relatively small proteins: the bluetongue virus membrane-penetration protein VP5 (EMD-6240/PDB 3J9E)[41], African swine fever virus major capsid protein p72 (EMD-0776/PDB 6KU9)[42] and C-terminal truncated human Pannexin1 (EMD-0975/PDB 6LTN)[43]. Among these, 3J9E displays an irregular shape composed of loops and alpha helices, while 6KU9 and 6LTN exhibit good structural symmetry with beta sheets and alpha helices, respectively. The predictions by RMSF-net exhibit strong agreement with the ground truth for these proteins, yielding correlation coefficients of 0.887, 0.731, and 0.757, respectively, as depicted in Fig. 3b. Predictions by RMSF-net_cryo and RMSF-net_pdb are supplied in Supplementary Figs. S1–S3. On 3J9E and 6KU9, both RMSF-net_cryo

and RMSF- net_pdb perform well, achieving correlations of 0.82, 0.69, and 0.881, 0.7, respectively. However, on 6LTN, RMSF- net_cryo only exhibits a correlation of 0.3 with the ground truth, possibly due to ring noise in the intermediate region of EMD-0975, leading to model misjudgment. In contrast, RMSF-net_pdb achieves a higher correlation of 0.767 on this protein, even surpassing RMSF-net, suggesting that instability factors in cryo-EM maps have a slight impact on RMSF-net's inference.

In addition to small proteins, Fig. 3c presents test examples of large protein complexes, including Mycobacterium tuberculosis RNA polymerase with Fidaxomicin[44] (EMD-4230/PDB 6FBV), RSC complex[45] (EMD-9905/PDB 6K15), and coronavirus spike glycoprotein trimer[46] (EMD-6516/PDB 3JCL). RMSF-net also excels on these complex structures, achieving correlation coefficients of 0.902, 0.819, and 0.804, respectively. Remarkably, these proteins are associated with human diseases and drug development, emphasizing the potential value of RMSF-net in facilitating drug development efforts. Predictions by RMSF-net_cryo and RMSF-net_pdb for these proteins are provided in Supplementary Figs. S4–S6, with correlation coefficients of 0.759 and 0.859 for 6FBV, 0.661 and 0.774 for 6K15, and 0.635 and 0.784 for 3JCL, respectively. Comparing model predictions in the Supplementary Figs. shows that RMSF-net aligns more closely with RMSF-net_pdb, supporting the previous argument that information from the PDB model plays a primary role in RMSF-net's feature processing.

We further applied RMSF-net to investigate dynamic changes in the NTCP protein during its involvement in biochemical processes. NTCP (Na+/taurocholate co-transporting polypeptide)[47] is a vital membrane transport protein predominantly located on the cell membrane of liver cells in the human body. It is responsible for transporting bile acids from the bloodstream into liver cells and plays a crucial role in the invasion and infection processes of liver viruses such as the hepatitis B virus. Therefore, structural and functional analysis of NTCP is crucial for liver disease treatment. In a previous study, two NTCP conformations, the inward-facing conformation (EMD-13593/PDB 7PQG) and the open pore conformation (EMD-13596/PDB 7PQQ), were resolved using cryo-electron microscopy. We performed dynamic predictions using RMSF-Net on these two conformations and revealed dynamic changes during the transition from the inward-facing to the open pore state, as shown in Fig. 3d. Compared to the inward-facing state, the open pore conformation displayed increased dynamics in the TM1 and TM6 regions of the panel domain, the TM7 and TM9 regions of the core domain, and the X motif in the center. Other regions maintained stability or exhibited enhanced stability. We hypothesize that the increased flexibility in these regions is associated with the relative motion between the panel and core domain in the open pore state, facilitating the transport of bile acids and the binding of preS1 of HBV in this conformation.

**Examples in more complex dynamic environments**

Despite exhibiting high-performance, RMSF-net is trained and tested based on relatively short-term (30 ns) MD simulations. In order to determine whether the structural fluctuation patterns obtained through the 30 ns simulation are stable enough for the model training, we performed longer simulations on three proteins[46,48,49] (PDB 3JCL, 6QNT, and 6SXA). The detailed setup and results are provided in the Supplementary Information (section "MD simulations over longer time periods"). In these cases, the results from the previous 30 ns simulation showed strong correlations with the results up to 500 ns, indicating that the 30 ns simulation can effectively capture stable structural fluctuation mode, thus qualified to serve as the foundation for our model training. The RMSF-net predictions also maintain high correlations with the long-term MD simulations, proving that the trained model has effectively absorbed the structural fluctuation patterns in MD simulations.

In addition, to make large-scale simulations feasible, we removed small molecules and ionic ligands during MD simulations, but ligands

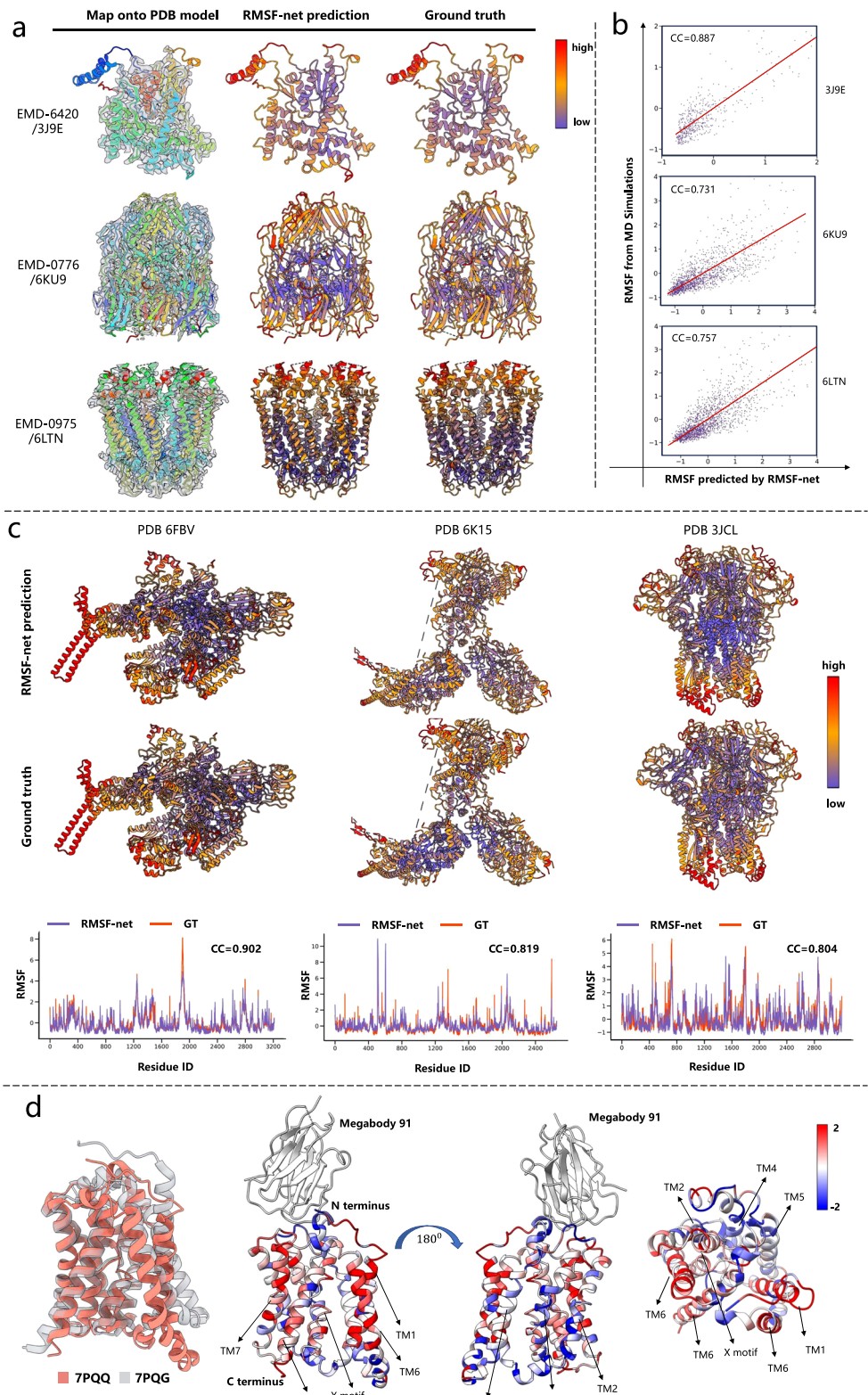

are included in the input density of RMSF-net. Therefore, the simulation results may be inaccurate regarding the flexibility of the ligand-containing protein, especially the flexibility near the ligands. To assess the impact of this treatment, we performed MD simulations for two protein systems containing ligands: the structure of the cargo bound AP-1:Arf1:tetherin-Nef closed trimer monomeric subunit[50] (EMD-7537/PDB 6CM9) and spastin hexamer in complex with substrate[51] (EMD-20226/PDB 6P07). Configurations of the MD simulations are provided

in the Supplementary Information (section "MD simulation configurations for ligand-binding proteins and membrane proteins"). The simulations with and without ligands exhibited high correlations in terms of RMSF, with correlations of 0.748 and 0.859 for the two proteins, respectively (Table 2). The predictions of RMSF-net also maintain comparable correlations with the additional simulations, of 0.757 and 0.859 respectively. This indicates that, overall, small molecule ligands have little impact on protein structural flexibility. However, we

**Fig. 3 | Examples of RMSF-net predictions. a, b** Show RMSF-net performance on three small proteins (EMD-6240/3J9E, EMD-0776/6KU9, EMD-0975/6LTN). **a** Visual comparison of RMSF-net predictions and ground truth. The first column shows the cryo-EM map overlaid on the PDB model. The second and third columns depict the PDB model colored according to the normalized RMSF values, using colors indicated by the color bar on the right. The second column represents the RMSF predictions by RMSF-net, and the third column represents the ground truth RMSF values from MD simulations. **b** Correlation plots between normalized RMSF-net predicted values and normalized ground truth values at residue levels for each protein. The normalized RMSF for residues is calculated as the average normalized RMSF of all atoms within that residue. Normalization is achieved by subtracting the mean and dividing by the standard deviation of RMSF in the PDB model. CC is an abbreviation for correlation coefficient. **c** RMSF-net performance on large protein complexes (PDB entry 6FBV, 6KU9, and 6LTN). The first and second rows display the PDB models of three protein complexes, with colors corresponding to the normalized RMSF values, indicated by the color bar on the right. The first row is colored based on the RMSF predictions by RMSF-net. The second row is colored according to the ground truth RMSF values from MD simulations. The third row demonstrates the profiles of normalized RMSF-net predicted values and normalized ground truth values along residue sequences for three proteins, where residue IDs correspond to the sequence order in the PDB models. CC is an abbreviation for correlation coefficient. **d** The dynamic change of NTCP protein from the inward-facing conformation to the open-pore conformation (PDB entry 7PQG/7PQQ). From left to right, the first cartoon illustrates the conformational transition of 7PQG to 7PQQ. The second, third, and fourth cartoons depict the dynamic changes in the conformational transition of this protein from the front, back, and top-down perspectives, respectively, where the RMSF difference is calculated and colored on the 7PQQ using the color bar provided in the upper right corner. The RMSF visualization was generated using PyMOL[59] and UCSF ChimeraX[60].

**Table 2 | The correlation between the RMSF of proteins obtained from MD simulation with ligands, without ligands, and RMSF-net**

| Data | MD sim. without ligands v.s. MD sim. with ligands | MD sim. without ligands v.s. RMSF-net prediction | MD sim. with ligands v.s. RMSF-net prediction |
|---|---|---|---|
| 6CM9 | 0.748 | 0.724 | 0.757 |
| 6P07 | 0.859 | 0.863 | 0.859 |

observed that near the ligands, the RMSF obtained from simulations without ligands is indeed higher than that obtained from simulations with ligands, as shown in Fig. 4. In these regions, the predicted values of RMSF-net are even closer to simulations with ligands, i.e., the predicted RMSF is lower. Our understanding of this phenomenon is that on one hand, the structure of the ligands is relatively small compared to the protein structure, so the scope of influence is limited and does not greatly affect the global distribution of structural flexibility. On the other hand, the local structures near the ligands became relaxed without the ligands in the original simulations. The deep model utilizes the learned pattern between protein internal structures and flexibility to infer the dynamics of the protein structure binding to the ligands. Although this is only an approximation, it has some correction effect.

Another aspect of MD simulation is the simulation of membrane proteins. In the sample preparation of cryo-EM, membrane proteins are purified and separated from membrane structures (Van Heel et al., 2000), which means that the structure and dynamics of membrane proteins in cryo-EM reflect their free state in solution. Correspondingly, our dynamic simulations were also performed in the membrane-free state. Therefore, our model is applicable to proteins in their free state in solution. However, in vivo, membrane proteins are attached to the cell membrane, so considering the simulation environment of the membrane will more accurately simulate their dynamics in biological systems. To explore the differences brought by the membranes, we conducted MD simulations in a membrane environment on two membrane proteins, the cryo-EM structure of TRPV5 (1-660) in nanodisc[52] (EMD-0593/PDB 6O1N) and cryo-EM structure of MscS channel, YnaI[53] (EMD-6805/PDB 5Y4O). The configurations of the MD simulations are provided in Supplementary Information (section "MD simulation configurations for ligand-binding proteins and membrane proteins"). The results, as shown in Table 3 and Fig. 5, demonstrate that the RMSF obtained from MD simulations with and without membranes maintain consistency overall, with correlations of 0.767 and 0.678 on 6O1N and 5Y4O respectively. The correlations between RMSF-net predictions and MD simulations with membranes are 0.804 and 0.675 respectively for these two proteins. As shown in Fig. 5b, the presence of the membrane leads to some changes in the flexibility of 5Y4O: in the upper region of 5Y4O, the RMSF obtained from MD simulations with the membrane is lower than that from MD simulations without membrane and RMSF-net. We speculate that this region may be influenced by membrane constraints, resulting in decreased flexibility, but the

overall flexibility distribution remains largely unchanged. In addition, we observe that on these two highly symmetrical structures, the predictions of RMSF-net also maintain symmetry similar to MD simulations.

## Model performance across different resolutions

The experimental results also indicate that our method exhibits consistent performance across cryo-EM maps with varying resolutions. The resolution of cryo-EM maps signifies the minimum scale at which credible structures are discernible within the map. In our dataset, there are more maps in the resolution range of 3–4 Å compared to 2–3 Å, as shown in Fig. 6a. Considering that our method takes cryo-EM maps of various resolutions into network training, concerns arise regarding potential model bias towards specific map resolutions. To address this concern, we conducted an analysis of the test performance of RMSF-net_cryo, Occ2RMSF-net, and RMSF-net compared to RMSF-net_pdb on maps of different resolutions. Results demonstrate that these models exhibit no significant performance differences across different resolution ranges, as shown in Fig. 6b–d. Only a minor deviation is observed in the range of 2–2.5 Å, which is statistically insignificant due to the limited number of 7 data points. This underscores that neural networks can fit data indiscriminately within the high-resolution range of 2–4 Å, without the need to process the maps to a uniform resolution at the preprocessing stage. The similar distributions of RMSF-net and Occ2RMSF-net across different resolutions, shown in Fig. 6b, e, further support the conclusion that dynamic inference from cryo-EM maps relies on an intermediate process of structural resolution. Furthermore, Fig. 6d demonstrates that, on average, RMSF-net outperforms RMSF-net_pdb across different resolution ranges, indicating that cryo-EM maps have an auxiliary effect on PDB models for dynamic analysis across different resolutions.

## Runtime assessment

In addition to its superior performance, RMSF-net demonstrates rapid inference capabilities and minimal storage overhead, whether running on high-performance GPUs or CPUs. Using a computer equipped with 10 CPU cores and 2 NVIDIA Tesla A100 GPUs, we conducted runtime assessments on the dataset, revealing a strong linear relationship between the execution time of RMSF-net and the data size. Moreover, compared to conventional MD simulations and DEFMap, this approach achieves substantial acceleration in processing speed.

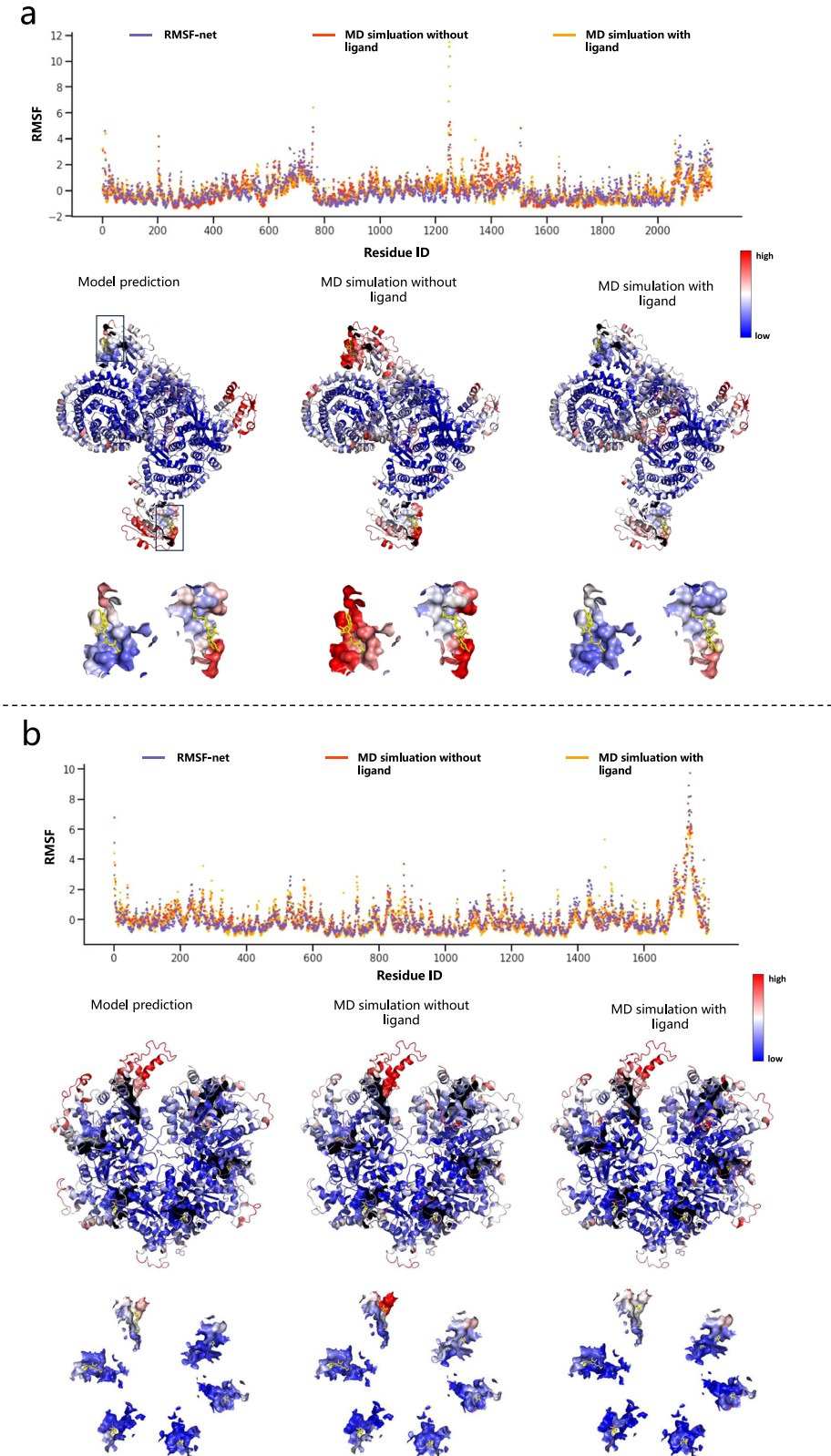

**Fig. 4 | Comparisons of RMSF obtained from MD simulations with ligands, without ligands, and RMSF-net on ligand-containing protein systems. a** Results for 6CM9. The top panel shows scatter plots of RMSF on residues, with colors corresponding to the three approaches as indicated in the legend. The middle panels present the visualizations of RMSF from the three approaches on the PDB structure, with colors corresponding to the normalized RMSF values, indicated by the color bar on the right. Ligands (GTP) are shown as yellow sticks, residues within 5 Å of the ligands are shown as surfaces, and black boxes indicate their positions. The bottom panels display the RMSF of structures near the ligands separately,

highlighting regions within the black boxes in the middle panels. **b** Results for 6P07. The top panel shows scatter plots of RMSF on residues, with colors corresponding to the three approaches as indicated in the legend. The middle panels present the visualizations of RMSF from the three approaches on the PDB structure, with colors corresponding to the normalized RMSF values, indicated by the color bar on the right. Ligands (ADP, ATP) are shown as yellow sticks, and residues within 5 Å of the ligands are shown as surfaces. The bottom panels display the RMSF of structures near the ligands separately.

**Table 3 | The correlation between the RMSF of proteins obtained from MD simulation with membrane, without membrane, and RMSF-net**

| Data | MD sim. without membrane v.s. MD sim. with membrane | MD sim. without membrane v.s. RMSF-net prediction | MD sim. with membrane v.s. RMSF-net prediction |
|---|---|---|---|
| 6OIN | 0.767 | 0.791 | 0.804 |
| 5Y4O | 0.678 | 0.737 | 0.675 |

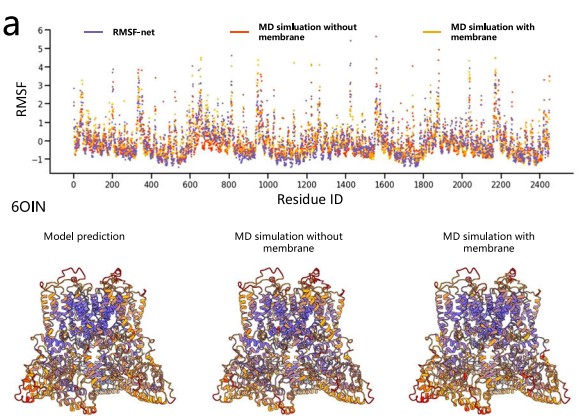
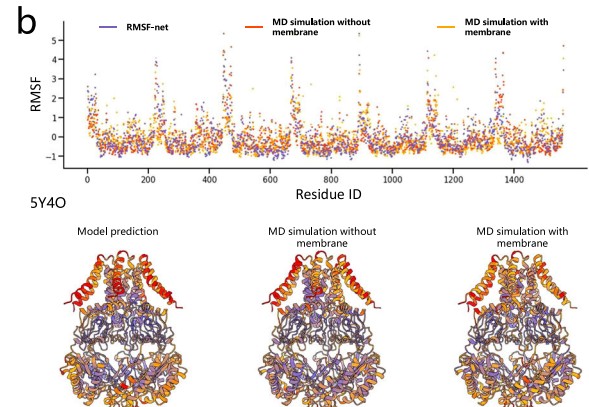

**Fig. 5 | Comparisons of RMSF obtained from MD simulations with membrane, MD simulations without membrane, and RMSF-net on membrane proteins. a** Results for 6O1N. **b** Results for 5Y4O. The top panels show scatter plots of RMSF on residues, with colors corresponding to the three approaches indicated in the legend. The bottom panels present the visualizations of RMSF from the three approaches on the PDB structure, with colors corresponding to the normalized RMSF values, indicated by the color bar in the middle.

As shown in Fig. 6e, Supplementary Figs. S11c and S12a, when executed on CPUs, RMSF-net's runtime is directly proportional to the weighted summation of cryo-EM map size and PDB model size, with a weight ratio of map size to PDB size at 0.0015:0.9985, both measured in units of k voxels. For most data, the weighted sum of map size and PDB size is within 500 k voxels, and processing can be completed in under a minute. When executed on GPUs, since most of the time is spent on preprocessing, the total time is linearly related to the map size, as shown in Fig. 6f, Supplementary Figs. S11a and S12b. For most maps with sizes below $300^3$ voxels, computations can be completed within 30 s. Detailed information regarding the RMSF-net processing time is provided in the Supplementary Information (section "Details of the RMSF-net processing time").

For comparative analysis, we selected ten relatively small maps from our dataset and performed runtime assessments using RMSF-net and DEFMap. As presented in Supplementary Tables S1–S3, across these ten data points, DEFMap exhibited processing times of 45.94 ± 31.84 minutes on CPUs and 37.51 ± 10.51 minutes on GPUs, concurrently generating data files of size 11.98 ± 8.30 GB. In contrast, RMSF-net showcased remarkable efficiency, with runtime durations of 16.66 ± 9.60 s and 3.09 ± 1.45 s on CPUs and GPUs, respectively, and yielding data files of 66.30 ± 31.06 MB. Both in terms of storage occupancy and time consumption, RMSF-net demonstrates significant improvements over DEFMap. Furthermore, in contrast to extended MD simulations, which often require hours or even days to perform simulations of 30 ns on individual proteins, RMSF-net delivers predictions with an average correlation of up to 0.75 and saves time and resources significantly, making it an ultra-fast means of performing protein dynamic analysis.

## Discussion

To address the issues of data insufficiency, model in need of optimization, and lack of interpretability in prior research, this study introduces RMSF-net, a deep learning-based protein dynamics prediction model. RMSF-net leverages the Unet++ architecture from three-

dimensional image segmentation and a regression module, employing dual features extracted from experimental cryo-EM maps and corresponding PDB models to infer atomic fluctuations in proteins. Through supervised training and evaluation on a high-resolution MD simulation dataset comprising 335 proteins, RMSF-net demonstrates superior performance compared to previous approaches, significantly improving data throughput. Rigorous five-fold cross-validation reveals that RMSF-net achieves an average test correlation coefficient of 0.746 on the dataset, with approximately 80% of the data exhibiting test correlations exceeding 0.7. This approach proves to be effective for both small protein and large protein complex examples, displaying robustness against noise and alignment errors in cryo-EM maps. Furthermore, runtime assessments on both CPUs and GPUs highlight RMSF-net's remarkable processing speed, transforming protein dynamic analysis from taking days and hours to mere minutes and seconds. The favorable performance of RMSF-net suggests its potential as an efficient substitute for large-scale MD simulations to elucidate macromolecular properties and physiological mechanisms and holds promise for practical applications in drug development, disease treatment, and other relevant fields.

We have further enhanced the interpretability of the dynamic prediction of RMSF-net through comparative experiments. By dividing the RMSF prediction process based solely on cryo-EM maps into two steps (Occ2RMSF-net), we confirmed that dynamics predictions by cryo-EM map-based models like DEFMap or RMSF-net_cryo are primarily achieved by deciphering protein structures. This highlights the connection between protein topology structure and dynamics, in accordance with the first principles of structure-function relationships. In addition, through a thorough comparison between RMSF-net_cryo, RMSF-net_pdb, and the final dual-combined RMSF-net, we have demonstrated that, on the one hand, the structure information from PDB models plays the primary role in RMSF-net, where the deep model learns patterns between structural topology and flexibility from MD simulations, and on the other hand, the dynamic information contained in the heterogeneous density distribution of cryo-EM maps

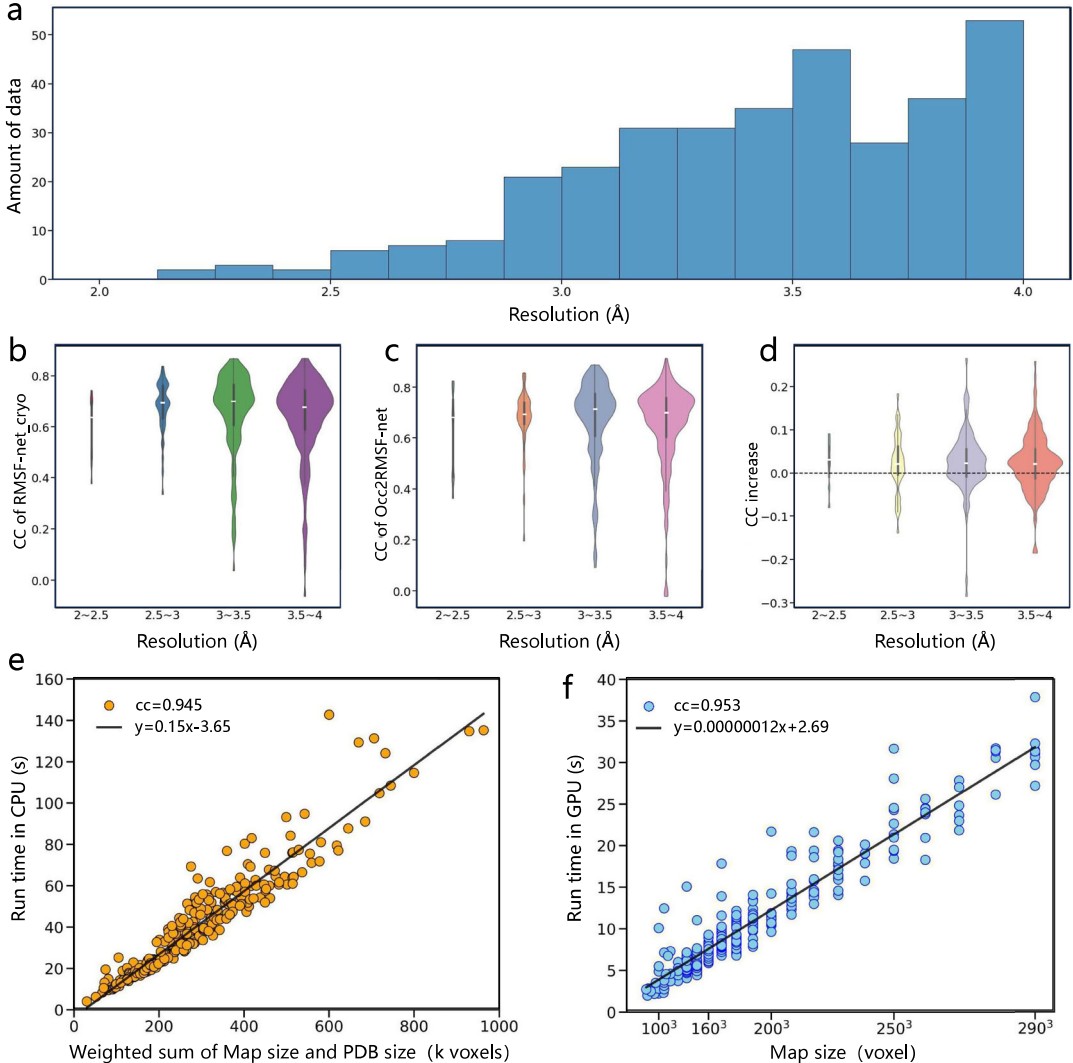

**Fig. 6 | The performance of the models at different map resolutions and the running efficiency of RMSF-net. a** Resolution distribution of cryo-EM maps in the dataset. **b–d** Shows the performance of RMSF prediction methods on different resolution maps in the dataset. For resolution groups "2–2.5", "2.5–3", "3–3.5" and "3.5–4", the sample sizes $N = 7, 42, 120,$ and 166. **b** Distribution of correlation coefficients (CC) of RMSF-net_cryo on maps across the four resolution ranges. **c** Distribution of correlation coefficients of Occ2RMSF-net across the four resolution ranges. **d** Distribution of correlation difference between RMSF-net and RMSF-net_pdb on the four resolution ranges. In **b–d**, the center white line, the lower and upper bound of the box in each violin plot indicate the median, the first quartile (Q1), and the third quartile (Q3), respectively. The whiskers of the boxes indicate Q1-1.5*IQR and Q3 + 1.5*IQR, with IQR representing the interquartile range.

The bounds of the violin plots show the minima and maxima, and the width indicates the density of the data. **e, f** Show the relationship between RMSF-net run time and data size. **e** The relationship between the RMSF-net run time on CPUs and the weighted sum of map size and PDB size among data points of the dataset. The map size and PDB size are weighted as 0.0015:0.9985 from linear regression, both taking k voxels as the units. The weighted sum range is set below 1000, which encompasses the majority of the data. The full range is presented in Supplementary Fig. S12a. **f** The relationship between the RMSF-net run time on GPUs and the map size among data points of the dataset. The map size is set below $300^3$, which encompasses the majority of the data. The full range is presented in Supplementary Fig. S12b.

further enhances the model. These results validate the complementary role of information from the cryo-EM map and PDB model for protein dynamic prediction in RMSF-net.

We have developed RMSF-net methods of three distinct schemes: the only cryo-EM map-based, the only PDB-model based and the dual-combined RMSF-net, and demonstrated their respective performance, making these methods potential to be applied to various scenarios, whether applied to freshly reconstructed cryo-EM maps, PDB structures derived from other structural elucidation techniques such as X-ray crystallography, or situations involving both cryo-EM maps and their fitted PDB structures. Specifically, with the rapid development of automated PDB model building tools in cryo-EM field[17,54–58], obtaining accurately calibrated PDB models of the cryo-EM density maps will no longer be difficult. The joint

analysis of the cryo-EM map and PDB model could be a new paradigm. Accurate flexibility prediction by RMSF-net can give more insight into the structure and behavior of proteins, thus assisting in relevant structural and functional tasks. For instance, when dealing with a recently reconstructed protein cryo-EM map, the dynamic prediction enables the assessment of flexibility in different regions, facilitating a more focused consideration of dynamically stable regions during structure modeling while imposing fewer constraints on the placement of structure domains in flexible areas. For proteins involved in physiological activities, RMSF-net enables the assessment of flexibility in distinct regions, identifying areas prone to conformational changes and establishing their functional relevance. For the convenient usage of RMSF-net, we have developed it into a software with a user-friendly web front-end built on the Django

framework. The detailed information about this application is supplied in the Supplementary Information (section "Accessible and user-friendly RMSF-net").

Admittedly, RMSF-net is mainly limited to predicting the flexibility of pure proteins and their complexes in solutions. For the dynamic nature of proteins in situations involving their binding to small molecule ligands or in membrane environments, this method may exhibit inaccuracies in some localized regions. While we have discussed cases where RMSF-net predictions still maintain some approximation, developing specialized methods for these cases is important. Given the high cost of accurately constructing force fields for small molecule ligands and simulating proteins on membrane environments, more data accumulation is needed. The superior performance of the RMSF-net reveals the feasibility of further research in this direction. Our study has also not yet expanded to nucleic acids and protein-nucleic acid complexes. Comprehensive characterization of various aspects in macromolecular dynamics, including multi-conformational prediction and transition analysis, requires further extensive and in-depth research in the future. Nevertheless, as a tool for predicting protein dynamics, RMSF-net remains promising for protein structure and dynamics studies due to its superior performance and ultra-fast processing speed.

### Reporting summary

Further information on research design is available in the Nature Portfolio Reporting Summary linked to this article.

## Data availability

The data that support this study are available from the corresponding authors upon request. The data used as examples are available from the EMDB and PDB (accession codes specified in the main text and figure captions). The source data underlying Figs. 2, 6, and Supplementary Figs. S11, S12 are provided as Source Data files. The full list of the dataset can also be found in the Source Data files. The MD simulation annotations for RMSF-net training and evaluation are available for download from Figshare (https://doi.org/10.6084/m9.figshare.25893670). The raw molecular dynamics trajectory data in this study are available from the corresponding authors upon reasonable request. Source data are provided with this paper.

## Code availability

The RMSF-net package is freely available for academic use [https://github.com/XintSong/RMSF-net].

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

## Acknowledgements

This research was supported by the National Key Research and Development Program of China [2021YFF0704300 to R.H.], the National Natural Science Foundation of China Projects Grant [62072280 to R.H., 32241027 and 61932018 to F. Z.], the King Abdullah University of Science and Technology (KAUST) Office of Research Administration (ORA) [Award No. REI/1/5234-01-01, REI/1/5414-01-01, RGC/3/4816-01-01, REI/1/5289-01-01, REI/1/5992-01-01, and URF/1/4663-01-0 to X.G], the Natural Science Foundation of Shandong Province [ZR2023YQ057 to R.H.], the Natural Science Foundation of Ningxia Province [2023AAC05036 to C.F.], and the Instrument Improvement Funds of Shandong University Public Technology Platform [ts20230204 to R.H.].

## Author contributions

R.H. and X.G. conceived the project and supervised the research. X.S. developed the methodology, performed the experiments, and analyzed the data. L.B. conducted molecular dynamics simulations. X.S., L.B., and F.Z. organized and wrote the paper. C.F. and Q.H. helped to revise the paper and provide scientific discussion when this study encountered problems.

## Competing interests

The authors declare no competing interests.
