## [Peer Review File · Nature Communications]

RMSF-net: Accurate Prediction of Protein Structural Flexibility by Integrating Intricate Atomic Structures and Cryo-EM Density InformationREVIEWER COMMENTS

Reviewer #1 (Remarks to the Author):

In this paper, the authors have developed a neural network model that rapidly predicts protein dynamics (computed by MD simulation) from cryo-EM density map and/or protein structure information. The number of the training dataset used in this study (335 protein structures) is much higher compared to those of previous methods, showing superior performance of this model. Although some of these results appear to contain previously unreported findings, there are several fundamental and technical concerns about the manuscript in its present form, as follows.

<Overall comments>

The protein flexibility (RMSF) has been computationally estimated by using Molecular Dynamics (MD) simulations, which require extensive computational cost. However, recent attention has shifted to the development of more accessible methodologies using AI. In comparison to an existing AI that predicts RMSF (Defmap), RMSF-net developed by the authors stands out for improving both prediction accuracy and computational speed, marking a significant advancement in the field of life sciences. Additionally, the identification of protein topology (3D structure) as a crucial factor in estimating structural flexibility adds a certain level of novelty. However, considering that RMSF-net is essentially an extension and refinement of the principles of Defmap, with expanded training data and improved algorithms, it could be viewed as a study with low originality. Therefore, I think the manuscript is more suitable for a specialized journal rather than a general one.

<Specific comments>

(i) In prediction of the flexibility of membrane proteins (e.g., NTLF) using RMSF-net, the methods section appears to lack clarification regarding whether the input data of the density map of proteins in the membrane environment include only the protein moieties or both the protein and membrane molecules. Also, I could not find details about the MD simulations of membrane proteins, such as whether they were conducted in the membrane or aqueous phase. Due to these omissions, I could not evaluate the consistency between the input data (density map obtained experimentally) and the predicted values (RMSF).

(ii) In RMSF-net implementation, the authors normalized the voxel densities within the boxes to a range of 0 to 1. However, the manuscript lacks detailed information of this process. When comparing the normalized intensities of boxes densely filled with protein atoms with those dominantly containing solvent molecules, the protein regions would have disproportionately higher intensities, suggesting a possibility of an unfair representation of features. The authors should include a description regarding this concern.

(iii) Although recognition of small molecules plays a crucial role in protein function, the authors conducted MD simulations after removing bound ligands observed in electron microscopy structures. Thus, RMSF-net's output can be regarded as RMSF of the apo-protein. Biologically-significant structural dynamics are expected to reside in RMSF of a protein in complex with its ligands or the difference in RMSF between the complex and the apo-protein, leading to offer information for protein functions and small molecule drug discovery strategies. In addition, while the RMSF values computed from 30 ns MD simulations (output of RMSF-net) represent nanosecond-timescale structural fluctuations, many of protein motions associated with their functions (e.g., open-closed conformational changes) occur on timescales beyond microseconds. In the manuscript, I found several statements regarding biological significance of RMSF-net's output (e.g., lines 365-366), however it seems unlikely that the nanosecond-timescale dynamics of apo-proteins provide information directly relevant to protein function or drug discovery.

(iv) The authors used combination of cryo-EM maps and PDB simulated maps as the input data, however these features are not independent but have a redundancy. The PDB simulated map is a

density map calculated reversely from a protein structure that is fitted to the cryo-EM map. Therefore, the PDB simulated map can be regarded as an idealized density map, which is generated through human intervention to eliminate noise and ambiguity attributed to partial flexibility. As the authors mentioned, the fact that a certain level of prediction accuracy was attained when using only the PDB simulated map as input suggests the significance of structural information in RMSF prediction. This prompts a question about necessity of information of the density. While the authors showed that inclusion of raw cryo-EM maps in the input data marginally improved the performance, I did not find an explicit clarification on additional factors contributing to the enhanced prediction accuracy.

Reviewer #2 (Remarks to the Author):

In the manuscript "RMSF-net: Accurate Prediction of Protein Structure Flexibility by Integrating Intricate Atomic Structures and Cryo-EM Density Information", the authors conducted comprehensive research on protein flexibility prediction based on cryo-EM structures. The dynamics of proteins play a crucial role in their structural and functional studies, and mainstream methods such as molecular dynamics simulations are time-consuming and resource-intensive. Cryo-electron microscopy stands out as an experimental technique for resolving protein structures, potentially containing information of multiple conformations and dynamics due to the characteristics of sample collection. The authors try to utilize the information underlying cryo-EM structures to infer the dynamic information of the target structure.

Methodologically, they utilized a convolutional network to predict RMSF (Root Mean Square Fluctuations) reflecting protein structural flexibility by simultaneously inputting PDB structures and cryo-EM maps in the form of three-dimensional images. In comparison to previous methods, the test results demonstrated a higher correlation with MD, reflecting the advantages of efficiency and low resource consumption. In the experimental part, they demonstrated the advantages of feature fusion of PDB and cryo-EM map over single feature through a series of comparative experiments, and also elucidated the decisive role of structural feature extraction for flexible prediction in the feature processing of neural networks by designing a two-phase approach and comparing it with the one-phase approach. The related platform is well-developed and easy to use.

I suggest the publication based on the condition that the following comments and suggestions are solved by the authors:

p.3: The subscripts i and j in Eq. 1 are not clear, and the meaning of \tilde{x} should be explained in the descriptions below.

Section 2.4: The first paragraph is not particularly relevant to the heading "RMSF-net training".

p.9 lines 9-11: The causal relationship in the statement, "The division was based on the maps rather than the segmented boxes, ... due to severe overfitting" is confusing.

p.9 lines 13: "During testing, the correlation between the predicted RMSF on the voxels...is computed as the evaluation metric" and the subsequent sentence "The correlation coefficients were computed at two levels: voxel level..." exhibit redundancy.

p.10 line -4: "the other parts" is unclear here; it should be clarified that it refers to other network components besides the input.

The last item in Table 1, 0.0097, is an order of magnitude smaller than the values for the std items in the table, which is likely to be an error. Verify if this is a statistical or publication error.

The term "normalized" in the captions of Figure 4 and Figure 5 is not explicitly explained.

Figures 8a, b, c show some redundancy with Figures 8d, e, f.

p.19 lines 2 : What does "without the need for a uniform resolution (as a hyperparameter)" specifically refer to?

p.19 Section 3.6 : The last sentence in the first paragraph "Detailed information regarding RMSF-net's processing time is provided in the Supplementary materials." is more appropriate at the end of the next paragraph.

Supplementary Figure 9 : Adding "detailed" or "details" to the first sentence of the caption would be more appropriate.

p.21 line -3 : "For a protein involved in". This is a general statement, so the plural form of "proteins" should be used.

On both Supplementary Table 1 and Supplementary Table 2, the "preprocessing" time for the first data are much longer than the other data. This may be due to the fact that the program's warm-up takes up a certain amount of time, and if so, measures should be taken to eliminate that factor.

Response to the reviewers

We sincerely thank the reviewers for their insightful and constructive feedback. We believe that the reviewers' comments have greatly improved our manuscript in terms of writing standards, details of data processing, and explanations of key technical and theoretical issues. Based on the reviewers' comments, we have made extensive modifications to the manuscript's structure, format, and scientific resulting presentation. In the revised manuscript, all modifications have been indicated using either red text or strike-through. The reviewers specifically raised some concerns regarding the novelty and technical rationale of our work. In response, we have restructured the paragraphs and sections of the article, and discussed the broad novelty of our work more clearly in the appropriate context of the "Introduction" and "Discussion" sections. Additionally, the reviewers expressed concerns about several technical issues. In response, we address these questions with new experimental results and more detailed explanations.

Below, please find the point-by-point response to all the reviewers' comments, where the number index indicates the original comments (in blue) and "Ans:" indicates our answers (in black). The changes in the manuscript and Supplementary Material are marked in red.

Reviewer 1

Remarks to the Author

In this paper, the authors have developed a neural network model that rapidly predicts protein dynamics (computed by MD simulation) from cryo-EM density map and/or protein structure information. The number of the training dataset used in this study (335 protein structures) is much higher compared to those of previous methods, showing the superior performance of this model. Although some of these results appear to contain previously unreported findings, there are several fundamental and technical concerns about the manuscript in its present form, as follows.

Ans: They you providing us with insightful, critical, and constructive feedback. These valuable comments helped us expand the research and significantly improved our manuscript.

We apologize for the unclear descriptions and improper organization of some sections in the manuscript, which may make some expressions unclear and mislead the readers. Thus, we expanded experiments and added relevant content to the revised manuscript and Supplementary Materials to address the technique details and related discussions. We also adjusted the paper chapters for more fluent reading. We have consolidated all the related contexts into Section 2.5 "Development of the RMSF-net method" to better clarify the technique details and benefits of the proposed model. Within Section 2.5, the contents of model results are arranged in the following order: Section 2.5.1 'Baselines', Section 2.5.2 'Enhancing interpretability of dynamics inferred from cryo-EM intensity', and Section 2.5.3 'Performance of RMSF-net that incorporates PDB information'. We have merged the contexts of model validation and evaluation from the original Section 2.5 into Section 2.4 and renamed this section "RMSF-net training and validation". In the "Introduction" and "Conclusion" sections, we specifically discussed the ideas, developments, interpretability, and applications of the methods in this work, and clarified the limitations of this study.

Please find our point-to-point response below to your comments.

Overall comments

The protein flexibility (RMSF) has been computationally estimated by using Molecular Dynamics (MD) simulations, which require extensive computational cost. However, recent attention has shifted to the development of more accessible methodologies using AI. In comparison to an existing AI that predicts RMSF (Defmap), RMSF-net developed by the authors stands out for improving both prediction accuracy and computational speed, marking a significant advancement in the field of life sciences. Additionally, the identification of protein topology (3D structure) as a crucial factor in estimating structural flexibility adds a certain level of novelty. However, considering that RMSF-net is essentially an extension and refinement of the principles of Defmap, with expanded training data and improved algorithms, it could be viewed as a study with low originality. Therefore, I think the manuscript is more suitable for a specialized journal rather than a general one.

Ans: Thank you for the insightful comments. In recent years, a number of works published in Nat. Methods (Maddhuri Venkata Subramaniya *et al.*, 2019; Zhang *et al.*, 2022; Baek *et al.*, 2024; Terashi *et al.*, 2024), Nat. Commun. (He *et al.*, 2022), Nat. Biotechnol. (Li *et al.*, 2024) and Nature (Jumper *et al.*, 2021; Jamali *et al.*, 2024) demonstrate that the utilization of deep models for the structure recognition of cryo-EM maps, structure prediction from sequences, and cryo-EM model building that integrates both methods, which reflects the fast-growing demand for AI-aided cryo-EM structure analysis. These methods focus primarily on the prediction of static information from the cryo-EM maps and protein structures. In contrast, DEFMap (Matsumoto *et al.*, 2021) shows the potential to extract the dynamic information (i.e., RMSF) from cryo-EM maps using a deep learning model. However, the relationship between the static and the dynamic information of the cryo-EM map has not yet been well-studied. Specifically, the structural flexibility is related to conformational changes in protein structure and binding site analysis in drug design, which has attracted more and more attention.

In the DEFMap study, the authors proposed that the reason why neural networks can extract structure flexibility from cryo-EM maps is that cryo-EM maps contain image features in the voxel density (for example, local resolution) associated with multiple conformations and atomic fluctuations. However, our experiments demonstrate that the structural information of the macromolecular complexes plays a more critical role in deep-learning-based flexibility prediction, which is a novel finding that has not been previously reported. Therefore, we propose that our work not only reports a more accurate and rapid method than DEFMap, but also reveals new scientific findings related to this important problem.

Figure 1 demonstrates the research idea and development of our method. First, we implemented our baseline RMSF-net (RMSF-net_cryo) with cryo-EM map as the input and the predicted RMSF scores as the output. Then, by dividing the one-stage cryo-EM map-based RMSF prediction (RMSF-net_cryo) into two-stage (Occ2RMSF-net), we took the structure information extracted from the original cryo-EM map into the network to predict RMSF, resulting in improved performance. Here, we demonstrated that the introduction of structure information can improve RMSF prediction. Then, we incorporated the PDB model which contains more accurate structure information, into our network. Compared with RMSF-net_cryo and Occ2RMSF_net, the RMSF-net networks trained with accurate PDB information (i.e., RMSF-net_pdb and RMSF-net_pdb01) significantly improved. Here, the improved RMSF-net_pdb and RMSF-net_pdb01 performance demonstrates that the structure information is critical in RMSF prediction. Finally, we built the final RMSF-net, which accepts both the cryo-EM map and PDB model information as input and outputs the predicted RMSF scores. Our final solution achieved higher prediction accuracy (0.746 correlation coefficient) compared with the networks that utilized only the cryo-EM map or PDB model as input, which illustrates that both the cryo-EM density map and PDB model help the inference of structural flexibility and RMSF-net benefits from accepting these two sources

Figure 1: Different RMSF prediction methods and their performance. DEFMap in the left is from (Matsumoto *et al.*, 2021) and the other methods are from our work. The upper section demonstrates the data flow for each method, where these methods can be classified into ‘Cryo-EM map based’, ‘PDB model based’ and ‘Dual (Cryo-EM map & PDB model)’. The axis below shows the average correlation coefficient between each method’s prediction scores and the MD simulations. From left to right, the correlation coefficient goes higher and the performance of the methods gets better.

of information. We have explored the relationship between static structure and dynamic flexibility in the above experiments, and the results show that the introduction of accurate structure information is more critical and will further improve flexibility prediction compared with the local image features underlying the cryo-EM density map. These findings provide scientific insight that clarifies the question raised in previous works and identifies directions for future works on accelerating structural dynamic analysis through deep neural networks.

Another contribution of our work is that RMSF-net, combined with well-engineered implementations, achieves high accuracy and fast inference for the structural flexibility of cryo-EM maps, making our method a convenient and practical tool for the community. RMSF-net can output more reliable RMSF predictions with a cryo-EM density map as input. Specifically, with the rapid development of automated PDB model building tools in cryo-EM field (Jamali *et al.*, 2024; Li *et al.*, 2024; Lugmayr *et al.*, 2023; Lawson *et al.*, 2021; Zhang *et al.*, 2022; Terashi *et al.*, 2024), obtaining accurately calibrated PDB models of the cryo-EM density maps will no longer be difficult. The joint analysis of the cryo-EM map and PDB model could be a new paradigm. Comprehensive experiments demonstrate the high correlation between our results and the MD simulations. The advantages of fast processing speed and low resource consumption further ensure its practicality. Additionally, we have constructed a large MD dataset to support the extensive training and validation of our model, which will be available to the community as a baseline dataset for the further development of related works. As suggested by the perspectives of Cell (Fraser and Murcko, 2024), Nat. Methods (Lane, 2023) and Nat. Rev. Mol. Cell Biol. (Kuhlman and Bradley, 2019), solutions for static protein structures have made significant progress with the advancement of structure prediction methods and cryo-EM techniques but have also reached a bottleneck. Currently, the research focus of the field has shifted toward studying protein dynamics and conformations to better understand the role of proteins in biological processes. Through

the deep learning model, RMSF-net establishes a link between the static cryo-EM structure and dynamic information showing the promising ability of deep learning methods to facilitate the study of protein structural dynamics.

In summary, we propose that our work has made three important contributions: 1) we discovered, for the first time, that the structural information of the macromolecular complex is a crucial factor in estimating structural flexibility; 2) we proposed a much more accurate and rapid method than the state-of-the-art; and 3) we provided a large benchmark dataset for the future studies along this direction. Therefore, we propose that our work makes both important scientific and technological contributions, and fits the scope and general audience of Nature Communications.

Specific comments

(i) In prediction of the flexibility of membrane proteins (e.g., NTLN) using RMSF-net, the methods section appears to lack clarification regarding whether the input data of the density map of proteins in the membrane environment include only the protein moieties or both the protein and membrane molecules. Also, I could not find details about the MD simulations of membrane proteins, such as whether they were conducted in the membrane or aqueous phase. Due to these omissions, I could not evaluate the consistency between the input data (density map obtained experimentally) and the predicted values (RMSF).

Ans: Thank you for this question. In this work, protein MD simulations uniformly used water as the solvent, without considering the membrane environment. Additionally, the input density maps do not contain membrane components, as protein samples in cryo-EM are purified proteins devoid of membrane structural components (Van Heel *et al.*, 2000). Therefore, these two aspects are consistent. However, considering the membrane for membrane proteins in the simulation environment more accurately simulates their dynamics in biological systems. We conducted membrane MD simulations for two membrane proteins in the dataset to explore the differences caused by the membrane. The relevant results including a table (Table 3) and a figure (Figure 10), have been added to Section 3.3 of the manuscript:

Table 3: The correlation between the RMSF of proteins obtained from MD simulation with membrane, without membrane, and RMSF-net.

Data	MD sim. without membrane v.s. MD sim. with membrane	MD sim. without membrane v.s. RMSF-net prediction	MD sim. with membrane v.s. RMSF-net prediction
6OIN	0.767	0.791	0.804
5Y4O	0.678	0.737	0.675

Another aspect of MD simulation is the simulation of membrane proteins. In the sample preparation of cryo-EM, membrane proteins are purified and separated from membrane structures (Van Heel *et al.*, 2000), which means that the structure and dynamics of membrane proteins in cryo-EM reflect their free state in solution. Correspondingly, our dynamic simulations were also performed in the membrane-free state. Therefore, our model is applicable to proteins in their free state in solution. However, in vivo, membrane proteins are attached to the cell membrane, so considering the simulation environment of the membrane will more accurately simulate their dynamics in biological systems. To explore the differences brought by the membranes, we conducted MD simulations in a membrane environment on

Figure 10: Comparisons of RMSF obtained from MD simulations with membrane, MD simulations without membrane, and RMSF-net on membrane proteins. (a):Results for 6O1N. (b):Results for 5Y4O. The top panels show scatter plots of RMSF on residues, with colors corresponding to the three approaches indicated in the legend. The bottom panels present the visualizations of RMSF from the three approaches on the PDB structure, with colors corresponding to the normalized RMSF values, indicated by the color bar on the right.

two membrane proteins, the cryo-EM structure of TRPV5 (1-660) in nanodisc (Dang *et al.*, 2019) (PDB 6O1N) and cryo-EM structure of MscS channel, YnaI (Yu *et al.*, 2018) (PDB 5Y4O). The configurations of the MD simulations are provided in Supplementary Materials. The results, as shown in Table 3 and Figure 10, demonstrate that the RMSF obtained from MD simulations with and without membranes maintain consistency overall, with correlations of 0.767 and 0.678 on 6O1N and 5Y4O respectively. The correlations between RMSF-net predictions and MD simulations with membranes are 0.804 and 0.675 respectively for these two proteins. Figure 10 b shows that the presence of the membrane leads to some changes in the flexibility of 5Y4O: in the upper region of 5Y4O, the RMSF obtained from MD simulations with membrane is lower than that from MD simulations without membrane and RMSF-net. We speculate that this region may be influenced by membrane constraints, resulting in decreased flexibility, but the overall flexibility distribution remains largely unchanged. Additionally, we observe that on these two highly symmetrical structures, the predictions of RMSF-net also maintain symmetry similar to MD simulations. ”

The simulation details of the membrane are provided in the “MD simulation configurations for ligand-binding proteins and membrane proteins” section of the Supplementary Materials, as follows:

“In addition, we conducted simulations in the membrane environment for two membrane proteins (PDB 6O1N (Dang *et al.*, 2019) and PDB 5Y4O (Yu *et al.*, 2018)). For simplicity, they were embedded in pure POPC bilayers (POPC, being the most common plasma membrane phospholipid (Marrink *et al.*, 2019)) and solvated with neutralizing ions (0.15 M NaCl) in rectangular boxes using CHARMM-GUI (Jo *et al.*, 2008). The orientation of proteins with respect to the membrane was predicted using the PPM web server 2.0 (Lomize *et al.*, 2012). For each initial structure, MD simulation was performed using Gromacs 2022 package (Abraham *et al.*, 2015). AMBER ff14SB (Maier *et al.*, 2015) and lipid21 force field (Dickson *et al.*, 2022) were used to describe the interactions between protein structures and POPC lipid molecules. Except for the addition of position restraints to membranes during the equilibrium stages, all the other control parameters remained consistent with those of previous simulations in a

membrane-free environment. ”

These results indicate that while the flexibility of some structures may be influenced by the membrane, the flexibility in most regions does not differ significantly between the two environments. In fact, conducting MD simulations without a membrane environment and using them as training data is reasonable because cryo-EM structures represent their free-state conformations in solution. Therefore, the convergence of the model will not be affected. Of course, establishing specialized methods for predicting the dynamics of membrane proteins is crucial for membrane protein research, which requires the accumulation of more simulation data on membrane proteins. The efforts made by (Newport *et al.*, 2019) have made significant strides. In the future, we will further investigate and develop specialized methods in this direction. Our current approach aims to predict the flexibility of proteins in aqueous solution, which is the most common scenario. We have also added relevant discussion to the Conclusion section of the paper, as follows:

“ Admittedly, RMSF-net is mainly limited to predicting the flexibility of pure proteins and their complexes in solutions. For the dynamic nature of proteins in situations involving their binding to small molecule ligands or in membrane environments, our methods may exhibit inaccuracies in some localized regions. While we have discussed cases where RMSF-net predictions still maintain some approximation, developing specialized methods for these cases is important. Given the high cost of accurately constructing force fields for small molecule ligands and simulating proteins on membrane environments, more data accumulation is needed. The superior performance of the RMSF-net reveals the feasibility of further research in this direction. ”

(ii) In RMSF-net implementation, the authors normalized the voxel densities within the boxes to a range of 0 to 1. However, the manuscript lacks detailed information of this process. When comparing the normalized intensities of boxes densely filled with protein atoms with those dominantly containing solvent molecules, the protein regions would have disproportionately higher intensities, suggesting a possibility of an unfair representation of features. The authors should include a description regarding this concern.

Ans: Thank you for the excellent suggestion. In this work, voxel density normalization is achieved through the following process: within each box, any density value less than 0 is set to 0, and then divided by the maximum density value within the box, thereby scaling the voxel density to a range between 0 and 1. We have now explained this process in Section 2.1 “RMSF-net Implementation” of the main text, as follows:

“ Voxel density normalization is achieved through the following process: within each box, any density values less than 0 are set to 0, and then divided by the maximum density value within the box, thus scaling the voxel density to a range from 0 to 1. ”

As stated in the main text, the main purpose of this is to scale the density of different maps to the same scale. As illustrated in the two maps in Figure 2, differences in density scales exist between the maps, and normalizing them to the same range before inputting them into the network helps with network processing. This operation was also adopted by EMNUSS (He and Huang, 2021).

Regarding the “high intensity disproportionality” and “unfair feature representation” issues, first, the aforementioned density normalization scales the original density proportionally. Second, in cryo-EM maps, due to the small molecular weight of solvent molecules, i.e., water molecules, and their dispersiveness compared to protein structures (as shown in Figure 2, water molecules are spaced far apart in space), their density is significantly lower than that of the protein region. Additionally, since the model input boxes all contain protein atoms, and the input box size is large enough to be 40^3 , almost

Figure 2: Density of water molecules in the cryo-EM map. We selected two proteins from the dataset with relatively many water molecules, the cryo-EM structure of monomeric photosystem II from *Synechocystis* sp. PCC 6803 lacking the water-oxidation complex (Gisriel *et al.*, 2020) (PDB 6WJ6, 198 water molecules), corresponding to the upper row of Figure a, and the overall structure of SLC26A9 (Chi *et al.*, 2020) (PDB 7CH1, 76 water molecules), corresponding to the lower row of Figure b. Each row, from left to right, consists of the PDB model and cryo-EM map overlaid on the PDB model above two different contour levels. In the PDB model, the protein structure is depicted in the yellow cartoon illustration, while water molecules are shown as red spheres. The cryo-EM map is displayed above the two contour levels: the first is below the recommended contour level, and the second is equal to the recommended contour level (from EMDB). The square gray boxes on the maps are magnified to show the density around the water molecules (as the water molecules are exposed, their density is lower than the contour levels).

all the input boxes contain more protein atoms than water. Therefore, in any case, the protein structure remains the primary density feature, while the density of water molecules is relatively low.

(iii) Although recognition of small molecules plays a crucial role in protein function, the authors conducted MD simulations after removing bound ligands observed in electron microscopy structures. Thus, RMSF-net’s output can be regarded as RMSF of the apo-protein. Biologically-significant structural dynamics are expected to reside in RMSF of a protein in complex with its ligands or the difference in RMSF between the complex and the apo-protein, leading to offer information for protein functions and small molecule drug discovery strategies. In addition, while the RMSF values computed from 30 ns MD simulations (output of RMSF-net) represent nanosecond-timescale structural fluctuations, many of protein motions associated with their functions (e.g., open-closed conformational changes) occur on timescales beyond microseconds. In the manuscript, I found several statements regarding biological significance of RMSF-net’s output (e.g., lines 365-366), however it seems unlikely that the nanosecond-timescale dynamics of apo-proteins provide information directly relevant to protein function or drug discovery.

Ans: Thank you very much for the insightful comments. We would like to address the two points you raised separately.

(i) First, we would like to account for the reason why we exclude ligands from our MD simulations:

Our dataset does not include nucleic acids, so the ligands present are small molecules or ions. However, building separate models for ligands is costly and requires quantum chemical calculations, which are not conducive to automation and large-scale simulations. Therefore, in this work, we conducted MD simulations on protein structures without small-molecule ligands to achieve large-scale standardized simulations. Another related study, ATLAS (Vander Meersche *et al.*, 2024), adopted a similar approach.

Excluding ligands may lead to inaccuracies to a certain extent because the protein structures near the original ligand binding sites become relatively unconstrained, which may exhibit higher flexibility in simulations than in situations when ligands bind them. We conducted additional MD simulations considering ligands on two protein systems to explore the dynamic differences between proteins with and without bound ligands. The relevant results include a table (Table 2) and two figures (Figures 8 and 9), which have been added to Section 3.3 “Discussion on MD simulations” in the manuscript:

Table 2: The correlation between the RMSF of proteins obtained from MD simulation with ligands, without ligands, and RMSF-net.

Data	MD sim. without ligands v.s. MD sim. with ligands	MD sim. without ligands v.s. RMSF-net prediction	MD sim. with ligands v.s. RMSF-net prediction
6CM9	0.748	0.724	0.757
6P07	0.859	0.863	0.859

“Additionally, to make large-scale simulations feasible, we removed small molecules and ionic ligands during MD simulations, but ligands are included in the input density of RMSF-net. Therefore, the simulation results may be inaccurate regarding the flexibility of the ligand-containing protein, especially the flexibility near the ligands. To assess the impact of this treatment, we performed MD simulations for two protein systems containing ligands: the structure of the cargo bound AP-1:Arf1:tetherin-Nef closed trimer monomeric subunit (Morris *et al.*, 2018) (PDB 6CM9) and Spastin hexamer in complex with substrate (Jones *et al.*, 2020) (PDB 6P07). Configurations of the MD simulations are provided in the Supplementary Materials. The simulations with and without ligands exhibited high correlations in terms of RMSF, with correlations of 0.748 and 0.859 for the two proteins, respectively (Table 2). The predictions of RMSF-net also maintain comparable correlations with the additional simulations, of 0.757 and 0.859 respectively. This indicates that, overall, small molecule ligands have little impact on protein structural flexibility. However, we observed that near the ligands, the RMSF obtained from simulations without ligands is indeed higher than that obtained from simulations with ligands, as shown in Figures 8 and 9. In these regions, the predicted values of RMSF-net are even closer to simulations with ligands, i.e., the predicted RMSF is lower. Our understanding of this phenomenon is that on one hand, the structure of the ligands is relatively small compared to the protein structure, so the scope of influence is limited and does not greatly affect the global distribution of structural flexibility. On the other hand, the local structures near the ligands became relaxed without the ligands in the original simulations. The deep model utilizes the learned pattern between protein internal structures and flexibility to infer the dynamics of the protein structure binding to the ligands. Although this is only an approximation, it has some correction effect.”

The configurations of the MD simulations with ligands are provided in the “MD simulation configu-

Figure 8: Comparisons of RMSF obtained from MD simulations with ligands, without ligands, and RMSF-net on 6CM9. The top panel shows scatter plots of RMSF on residues, with colors corresponding to the three approaches as indicated in the legend. The middle panels present the visualizations of RMSF from the three approaches on the PDB structure, with colors corresponding to the normalized RMSF values, indicated by the color bar on the right. Ligands (GTP) are shown as yellow sticks, residues within 5 Å of the ligands are shown as surfaces, and black boxes indicate their positions. The bottom panels display the RMSF of structures near the ligands separately, highlighting regions within the black boxes in the middle panels.

rations for ligand-binding proteins and membrane proteins” section of the Supplementary Materials, as shown below:

“ We selected two ligand-binding proteins (PDB 6CM9 (Morris *et al.*, 2018) and PDB 6P07 (Jones *et al.*, 2020)) to evaluate the effect of ligands on protein flexibility. The parameters used for the protein structure were AMBER ff14SB force field (Maier *et al.*, 2015). The ATP, ADP and GTP molecule parameters were taken from the AMBER parameter database (Meagher *et al.*, 2003). All other protocols were consistent with the previous ligand-free protein simulations except for the addition of position restraints to ligands during the equilibrium stages. ”

The above results indicate that ligands have some impact on the flexibility of local structures near the ligands; however, they have little effect on overall protein flexibility. Since the regions near the ligands only account for a small part of the total structure, their impact on model convergence is minimal.

Figure 9: Comparisons of RMSF obtained from MD simulations with ligands, without ligands, and RMSF-net on 6P07. The top panel shows scatter plots of RMSF on residues, with colors corresponding to the three approaches as indicated in the legend. The middle panels present the visualizations of RMSF from the three approaches on the PDB structure, with colors corresponding to the normalized RMSF values, indicated by the color bar on the right. Ligands (ADP,ATP) are shown as yellow sticks, residues within 5 Å of the ligands are showed as surfaces. The bottom panels display the RMSF of structures near the ligands separately.

Additionally, the model's input includes ligands, meaning that the structures inputted into the model are in their bound state with the ligands; and the trained model treats the protein regions that bind to ligands as protein internal structures, which can achieve a certain approximation effect. Nevertheless, we acknowledge that our method is primarily aimed at predicting the flexibility of pure protein structures and their complexes (protein-protein binding). Our method still cannot guarantee accurate predictions of the dynamics of small-molecule ligand binding to proteins. We have added relevant discussion to the Conclusion section regarding the limitations on the applicability of this method, as follows:

“ Admittedly, RMSF-net is mainly limited to predicting the flexibility of pure proteins and their complexes in solutions. For the dynamic nature of proteins in situations involving their binding to small molecule ligands or in membrane environments, our methods may exhibit inaccuracies in some

localized regions. While we have discussed cases where RMSF-net predictions still maintain some approximation, developing specialized methods for these cases is important. Given the high cost of accurately constructing force fields for small molecule ligands and simulating proteins on membrane environments, more data accumulation is needed. The superior performance of the RMSF-net reveals the feasibility of further research in this direction. ”

Accurately studying the dynamics of protein binding to small-molecule ligands or nucleic acids is crucial. At present, efforts are also being made in this field to accumulate high-precision data, such as MISATO (Siebenmorgen *et al.*, 2023). We will also attempt to improve data quality while developing specialized methods.

(ii) We have also conducted additional experiments regarding the time scale of the MD simulations to clarify the reviewer’s concerns:

Regarding the gap between the structural flexibility obtained from 30 ns time-scale MD simulations and protein function (e.g., conformational changes such as opening and closing), we have implemented some further validation measures and conducted in-depth discussions.

First, we conducted additional simulations at longer time scales for several proteins, including one for 200 ns and two for 500 ns to ensure the stability of structural flexibility obtained from the 30 ns MD simulations used for model training. The detailed results, including three figures (Supplementary Figure 8~10), are provided in the “MD simulations over longer periods of time” section of the Supplementary Materials as follows:

Supplementary Figure 8: Comparisons of RMSF obtained from MD simulations for 30 ns, 200 ns, and RMSF-net on 3JCL. The first column shows scatter plots of RMSF on residues, with colors corresponding to the three approaches indicated in the legend. The second column presents the RMSF visualization from the three approaches on the PDB structure, with colors corresponding to the normalized RMSF values, indicated by the color bar on the right.

Supplementary Figure 9: Comparisons of RMSF obtained from the new several hundred nanosecond MD simulations, the original 30 ns MD simulation, and RMSF-net on 6QNT. The top panel shows the RMSF visualizations obtained from MD simulations ranging from 100 to 500 ns on the PDB structure, with colors corresponding to the normalized RMSF values, indicated by the color bar, as in Supplementary Figure 8. The bottom-left panel presents the visualizations of the original 30 ns simulation and the predicted values of RMSF-net. The correlation coefficient curves in the bottom-right panel outline the correlation between the original MD simulation and the new simulation, as well as the correlation between RMSF-net and the new MD simulation over time.

“ We conducted MD simulations for hundreds of nanoseconds on three proteins in the dataset to ascertain the robustness of the structural fluctuations obtained from the 30 ns simulation, including the cryo-EM structure of a coronavirus spike glycoprotein trimer (Walls *et al.*, 2016) (PDB 3JCL), the Human Adenovirus type 3 fiber knob in complex with one copy of Desmoglein-2 (Vassal-Stermann *et al.*, 2019) (PDB 6QNT), and the XPF-ERCC1 cryo-EM structure, apo-form (Jones *et al.*, 2020) (PDB 6SXA). We initially performed a 200 ns simulation for 3JCL, keeping all the settings the same as before except for extending the final production run time. Following the simulation, we computed the RMSF and compared it with the previous 30 ns simulation; the corresponding scatter plots and visualizations are provided in Supplementary Figure 8. The results reveal a high correlation of 0.811 between the new and the old simulations. Furthermore, the predicted values of RMSF-net exhibit a correlation of 0.819 with the 200 ns MD simulation, which is even higher than the correlation of 0.804 with the original 30 ns. Subsequently, we conducted simulations lasting up to 500 ns on 6QNT and 6SXA and computed the RMSF obtained for five simulation durations from 100 ns to 500 ns. The RMSF obtained from the previous 30 ns simulations show high correlations with the new simulations at all five

Supplementary Figure 10: Comparisons of RMSF obtained from the new several hundred nanosecond MD simulations, the original 30 ns MD simulation, and RMSF-net on 6SXA. The top panel shows the RMSF visualizations obtained from MD simulations ranging from 100 to 500 ns on the PDB structure, with colors corresponding to the normalized RMSF values, indicated by the color bar, as in Supplementary Figure 8. The bottom-left panel presents the visualizations of the original 30 ns simulation and the predicted values of RMSF-net. The correlation coefficient curves in the bottom-right panel outline the correlation between the original MD simulation and the new simulation, as well as the correlation between RMSF-net and the new MD simulation over time.

durations, as depicted in Supplementary Figures 9 and 10. The correlation remains at approximately 0.8 for 6QNT, while it consistently exceeds 0.85 for 6SXA; both show no evident decrease over time, indicating that the structural fluctuations obtained from the 30 ns simulation were sufficiently stable and can serve as the foundation for our model training. Additionally, the correlations between the predicted values of RMSF-net for both 6QNT and 6SXA and the MD simulations remain above 0.8 for all five durations, demonstrating that our model genuinely learns the structural fluctuation patterns from the MD simulations. ”

We have also added the relevant content to Section 3.3 “Discussion on MD simulations” of the main text as follows:

“ Despite exhibiting high performance, RMSF-net is trained and tested based on relatively short-term (30 ns) MD simulations. In order to determine whether the structural fluctuation patterns obtained through the 30 ns simulation are stable enough for the model training, we performed longer simulations on three proteins (PDB 3JCL (Walls *et al.*, 2016), 6QNT (Vassal-Stermann *et al.*, 2019), and 6SXA (Jones *et al.*, 2020)). The detailed setup and results are provided in the Supplementary Materials and Supplementary Figures 8-10. In these cases, the results from the previous 30 ns simulation showed strong correlations with the results up to 500 ns, indicating that the 30 ns simulation can effectively capture

stable structural fluctuation mode, thus qualified to serve as the foundation for our model training. The RMSF-net predictions also maintain high correlations with the long-term MD simulations, proving that the trained model has effectively absorbed the structural fluctuation patterns in MD simulations. ”

The above results indicate that 30 ns simulations can capture the patterns of atomic fluctuations well. Thus, their results are qualified to serve as the training data for the model. Additionally, the experimental results demonstrate that the trained model learns these patterns effectively.

Second, we have carefully identified the link between short/long timescale simulations and protein functions. It is generally believed that structural flexibility reflects high-frequency structural fluctuations. In contrast, large structural transitions occur on longer time scales and are directly related to functional execution. However, this does not mean that short-term MD simulations are irrelevant to actual biological functions. (Vander Meersche *et al.*, 2024) argued that “protein structure ensembles generated using MD trajectories of tens of nanoseconds enhance docking performance (17–21), allow detection of pockets participating in protein-protein interaction (22) or detect flexibility patterns characteristic for residues involved in protein-protein interface formation (23). MD simulations lasting for hundreds of nanoseconds allow detection of allosteric pathways (24–26), while longer MD can bring valuable insights on major conformational changes (27,28).” (Choy *et al.*, 2017) explored the relationship between short-term/long-term dynamics, and macromolecular function based on enzyme catalytic function, suggesting that fast time-scale motions combined with slower time-scale actions can jointly regulate the catalytic cycle, in agreement with the findings of (Henzler-Wildman and Kern, 2007) in a study done in 2007. The above studies suggest that the structural flexibility of a macromolecule in a single conformation actually implies the direction of the next step, and is, therefore, closely related to the conformational transitions and functional execution presented by long-term dynamics. In other words, the flexibility of a protein provides the basis for subsequent large conformational transitions, which together influence the structure and function of the protein. Additionally, it is important to clarify that the goal of RMSF-net is to predict the conformational flexibility of a protein structure, which corresponds to the flexibility of the structure represented by the cryo-EM map and its fitted PDB model. For a wider range of conformational transitions (e.g., “open-closed conformational transitions”), the RMSF-net provides a potential application that first predicts the flexibility of each individual conformation and then establishes a function-structure linkage based on the difference in flexibility between conformations. Therefore, as the training data for this method, achieving functional conformational changes (such as opening and closing conformational changes) through MD simulation is not necessary under current demands and requires complex physiological environments and significant computational overhead. In summary, we propose that the 30 ns simulation in this study is reasonable.

Based on the above discussions, we have revised the paragraph of the sentence you mentioned (lines 355-356), which is in the third paragraph in the “Conclusion” section, to conclude the methods in this work and noted some potential applications for these methods, as follows:

“ We have developed RMSF-net methods of three distinct schemes: the only cryo-EM map based, the only PDB-model based and the dual-combined RMSF-net, and demonstrated their respective performance, making these methods potential to be applied to various scenarios, whether applied to freshly reconstructed cryo-EM maps, PDB structures derived from other structural elucidation techniques such as X-ray crystallography, or situations involving both cryo-EM maps and their fitted PDB structures. Specifically, with the rapid development of automated PDB model building tools in cryo-EM field (Jamali *et al.*, 2024; Li *et al.*, 2024; Lugmayr *et al.*, 2023; Lawson *et al.*, 2021; Zhang *et al.*, 2022; Terashi *et al.*, 2024), obtaining accurately calibrated PDB models of the cryo-EM density maps will no longer be difficult. The joint analysis of the cryo-EM map and PDB model could be a new paradigm. Accu-

rate flexibility prediction by RMSF-net can give more insight on the structure and behavior of proteins, thus assisting in relevant structural and functional tasks. For instance, when dealing with a recently reconstructed protein cryo-EM map, the dynamic prediction enables the assessment of flexibility in different regions, facilitating a more focused consideration of dynamically stable regions during structure modeling while imposing fewer constraints on the placement of structure domains in flexible areas. For proteins involved in physiological activities, RMSF-net enables the assessment of flexibility in distinct regions, identifying areas prone to conformational changes and establishing their functional relevance. ”

(iv) The authors used combination of cryo-EM maps and PDB simulated maps as the input data, however these features are not independent but have a redundancy. The PDB simulated map is a density map calculated reversely from a protein structure that is fitted to the cryo-EM map. Therefore, the PDB simulated map can be regarded as an idealized density map, which is generated through human intervention to eliminate noise and ambiguity attributed to partial flexibility. As the authors mentioned, the fact that a certain level of prediction accuracy was attained when using only the PDB simulated map as input suggests the significance of structural information in RMSF prediction. This prompts a question about necessity of information of the density. While the authors showed that inclusion of raw cryo-EM maps in the input data marginally improved the performance, I did not find an explicit clarification on additional factors contributing to the enhanced prediction accuracy.

Ans: Thank you for identifying this. Here, we discuss the differences and connections between the PDB simulated map and cryo-EM map from two aspects:

1. The average static structure represented by the PDB model and PDB simulated map: The PDB model is constructed from the cryo-EM map, representing the most likely or average position of each atom in the cryo-EM map. Since the position of each atom is uniquely determined, it can be considered as a static structure representing the fundamental topology of the protein structure in that conformation. The simulated map is generated by reverse computation of the PDB model, and its density depends only on the spatial distribution of atoms. Therefore, predicting flexibility based on these maps in RMSF-net_pdb is considered a pattern between topology and flexibility. The reason for simulating the PDB model as a density map has two aspects: (1) this structured three-dimensional grid data structure is perfectly suited for CNNs, and (2) it facilitates integration with cryo-EM maps.

2. The dynamic information in the cryo-EM map. The density of the cryo-EM map is determined primarily by the spatial arrangement of protein atoms in the static state; however, it is also influenced by other factors, including noise, structural flexibility, etc., resulting in different resolutions in different regions of the cryo-EM map. Since structural flexibility can cause changes in the density distribution of the cryo-EM map, the cryo-EM map contains dynamic information that is not present in the PDB simulated map.

Besides, in DEFMap (Matsumoto *et al.*, 2021), the authors calculated the correlation between the density and local resolution of the cryo-EM map and RMSF on their dataset, with correlations of 0.459 ± 0.179 for 25 maps and 0.510 ± 0.091 for 15 maps (other 10 negative), respectively. Here, we illustrate the relationship between density and RMSF through the cryo-EM maps of 6QNT (Vassal-Stermann *et al.*, 2019) and 6SXA (Jones *et al.*, 2020), as shown in Figure 3. As the density threshold increases, the density of regions with higher flexibility become invisible first in the cryo-EM map, while the density invisibility is more uniform in the PDB simulated map. This indicates that the density is unrelated to the flexibility of the underlying structure in simulated maps, whereas regions with greater flexibility have

Figure 3: The density of protein local structures with different flexibilities in cryo-EM maps and PDB simulated maps. The upper row (a) depicts the situation on PDB 6QNT, while the lower row (b) illustrates PDB 6SXA. Each row, from left to right, displays the PDB model colored according to normalized RMSF (obtained from the molecular dynamics simulation in the manuscript, indicated by the color bar besides), the cryo-EM map overlaid on the colored PDB model at two contour levels, and the PDB simulated map overlaid on the colored PDB model at two contour levels. It can be observed that with increasing contour level, the density of regions with higher flexibility become invisible first in the cryo-EM map, while the density invisibility is more uniform in the PDB simulated map, showing no relation to flexibility of the underlying structure.

lower density in cryo-EM maps, which are relatively blurred. This is because the projections of multiple two-dimensional images in the flexible region do not overlap in three-dimensional reconstruction.

We propose that this dynamic-related characteristic of cryo-EM maps enhances the performance of RMSF-net compared to RMSF-net_pdb. The mentioned “redundancy” is actually the similarity between the cryo-EM map and the PDB simulated map we discussed in reflecting the static topological structure. However, we propose that this similarity has an “alignment” effect at the input end of the deep model, promoting the superior performance of RMSF-net. Another interesting point is that, there are average positions and instantaneous positions in the RMSF definition. The PDB model built from the cryo-EM map corresponds precisely to the average position of the structure. The cryo-EM map reconstructed from multiple particles in the sample corresponds to the information of multiple instantaneous conformations. Thus, in our analysis, the information provided by the PDB model can be compared to the mathematical expectation in a distribution function and the voxel densities of the cryo-EM map reconstructed from multiple instantaneous conformations can be compared to the mathematical variance of the distribution function. This heuristic idea prompted us to initially choose to integrate the PDB model and the cryo-EM map to infer RMSF through neural networks.

We have added relevant content in the main text. In the Introduction section, we add the following sentences where we discuss the DEFMap study (Matsumoto *et al.*, 2021):

“ In addition, the cryo-EM maps contain both static structure information of the protein and local density changes due to structural fluctuations, i.e., inhomogeneity of the density map due to flexibility. In DEFMap(Matsumoto *et al.*, 2021), the authors calculated the correlation coefficients of local density

and local resolution with RMSF. Both exceed 0.4 in more than half of the data. This indicates a correlation between the local density distribution of cryo-EM maps and structural flexibility. However, due to the black-box nature of neural networks, whether there is indeed a pattern between local density distribution and flexibility in the network and what role the static structure information plays in it are still unclear. ”

We have reorganized the section sequence in the manuscript to clarify the coherence and development among the methods. We consolidated the contents of all the models into Section 2.5 “Development of the RMSF-net method”. Below this section, the model results are arranged in the following order: Section 2.5.1 ‘Baselines’, Section 2.5.2 ‘Enhancing interpretability of dynamics inferred from cryo-EM intensity’, and Section 2.5.3 ‘Performance of RMSF-net that incorporates PDB information’. After the model results, we added the following discussion at the beginning of the “Anomalies” subsection (Section 3.1) in the “Discussions” section:

“ The experimental results above prove that the combination of cryo-EM map and PDB model results in the superior performance of RMSF-net. As shown in Figure 4 b, the prediction of RMSF-net is better on most cases comparing models utilizing only the cryo-EM map or only the PDB model. Because the PDB models are built from the corresponding cryo-EM maps, their spatial coordinates are naturally aligned and their structural information is consistent. Moreover, the PDB model built from the cryo-EM map corresponds precisely to the average position of the structure, and the cryo-EM map reconstructed from multiple particles in the sample corresponds to the information of multiple instantaneous conformations. By combining the ‘expectation’ and conformational ‘variance’ from the two sources, we believe that this structural consistency and complementarity create an alignment effect, and promote the superior performance of RMSF-net. ”

In the Conclusion section, we have revised the second paragraph, as follows:

“ We have further enhanced the interpretability of the dynamic prediction of RMSF-net through comparative experiments. By dividing the RMSF prediction process based solely on cryo-EM maps into two steps (Occ2RMSF-net), we confirmed that dynamics predictions by cryo-EM map-based models like DEFMap or RMSF-net_cryo are primarily achieved by deciphering protein structures. This highlights the connection between protein topology structure and dynamics, in accordance with the first principles of structure-function relationships. In addition, through a thorough comparison between RMSF-net_cryo, RMSF-net_pdb and the final dual-combined RMSF-net, we have demonstrated that on one hand, the structure information from PDB models plays the primary role in RMSF-net, where the deep model learns patterns between structural topology and flexibility from MD simulations. and on the other hand, the dynamic information contained in the heterogeneous density distribution of cryo-EM maps further enhances the model. These results validate the complementary role of information from cryo-EM map and PDB model for protein dynamic prediction in RMSF-net.”

Finally, we would like to sincerely thank the reviewer again for these insightful and constructive comments, which have motivated us to conduct more experiments and analyses, restructure and revise our manuscript, and greatly improved the quality of our work. We are truly grateful for your time and efforts.

Reviewer 2

Remarks to the Author

In the manuscript “RMSF-net: Accurate Prediction of Protein Structure Flexibility by Integrating Intricate Atomic Structures and Cryo-EM Density Information”, the authors conducted comprehensive research on protein flexibility prediction based on cryo-EM structures. The dynamics of proteins play a crucial role in their structural and functional studies, and mainstream methods such as molecular dynamics simulations are time-consuming and resource-intensive. Cryo-electron microscopy stands out as an experimental technique for resolving protein structures, potentially containing information of multiple conformations and dynamics due to the characteristics of sample collection. The authors try to utilize the information underlying cryo-EM structures to infer the dynamic information of the target structure.

Methodologically, they utilized a convolutional network to predict RMSF (Root Mean Square Fluctuations) reflecting protein structural flexibility by simultaneously inputting PDB structures and cryo-EM maps in the form of three-dimensional images. In comparison to previous methods, the test results demonstrated a higher correlation with MD, reflecting the advantages of efficiency and low resource consumption. In the experimental part, they demonstrated the advantages of feature fusion of PDB and cryo-EM map over single feature through a series of comparative experiments, and also elucidated the decisive role of structural feature extraction for flexible prediction in the feature processing of neural networks by designing a two-phase approach and comparing it with the one-phase approach. The related platform is well-developed and easy to use.

Ans: Thank you for the positive feedback and constructive comments. We followed all of them to improve the quality of the paper further. Please find the point-to-point response below to your comments.

I suggest the publication based on the condition that the following comments and suggestions are solved by the authors:

p.3: The subscripts i and j in Eq. 1 are not clear, and the meaning of \bar{x} should be explained in the descriptions below.

Ans: Thanks for the correction. We have updated Eq. 1 in the revised manuscript and provided the exact meaning of each symbol in the equation, as follows: “

$$\text{RMSF} = \sqrt{\frac{1}{T} \sum_{t=1}^T (x(t) - \bar{x})^2} \quad (1)$$

where x represents the real-time position of atoms or residues, t represents time and \bar{x} represents mean position over a period of time T .”

Section 2.4: The first paragraph is not particularly relevant to the heading “RMSF-net training”.

Ans: Thanks for the suggestion. We apologize for the improper organization of the section. The first paragraph in section 2.4 describes the annotation of RMSF on PDB atoms and cryo-EM map voxels, which is closely related to the training and validation of model. Therefore, we have merged the contents related to cross-validation and evaluation metrics from Section 2.5 into this section, and renamed this section (Section 2.4) to “RMSF-net training and validation”.

p.9 lines 9-11: The causal relationship in the statement, “The division was based on the maps rather than the segmented boxes, ... due to severe overfitting” is confusing.

Ans: Thanks for identifying the issue. We have revised the original text “The division was based on the maps rather than the segmented boxes, ... due to severe overfitting” to “**In particular, the division was based on the maps rather than the segmented boxes in order to ensure independence between these sets.**” to better explain the way we partitioned the dataset, which is in the penultimate paragraph of Section 2.4 of the revised manuscript.

p.9 line 13: “During testing, the correlation between the predicted RMSF on the voxels...is computed as the evaluation metric” and the subsequent sentence “The correlation coefficients were computed at two levels: voxel level...” exhibit redundancy.

Ans: Thanks for bringing this up. We find that there was indeed a repetition here. The phrase “on the voxels” in the first sentence is redundant, so we have removed it. In the revised manuscript, this part is placed at the end of Section 2.4, as follows:

“ During testing, the correlation coefficients between the predicted RMSF ~~on the voxels~~ and the ground truth (RMSF values derived from MD simulations) were computed as the evaluation metric. The correlation coefficients were computed at two levels: voxel level, corresponding to RMSF on the map voxels, and residue level, corresponding to RMSF on the PDB model residues (**obtained by averaging RMSF on the corresponding atoms**)...”

We first explained our use of correlation coefficients as measure for the model's performance, and then pointed out that we calculated the correlation coefficients at two levels.

p.10 line -4: “the other parts” is unclear here; it should be clarified that it refers to other network components besides the input.

Ans: Thanks for for the excellent suggestion. As you mentioned, “the other parts” here refers to the remaining network components apart from the input, or rather, the main body of the network. Therefore, we have replaced ‘the other parts’ with ‘the main part of the network’, as follows:

“ As outlined in the Method “**RMSF-net implementation**” section, RMSF-net takes the dual-channel feature of **density from the** cryo-EM map and PDB simulated map at the same spatial position as input, while ~~the other parts~~ **the main part of the network** remains the same as RMSF-net_cryo and RMSF-net_pdb. ”

The last item in Table 1, 0.0097, is an order of magnitude smaller than the values for the std items in the table, which is likely to be an error. Verify if this is a statistical or publication error.

Ans: Thanks for the correction. After careful checking, we found that there was a manual error in registering this data on Table 1, where an extra decimal place was added. The actual statistical result is **0.097** instead of 0.0097. We apologize for this mistake, and we have corrected this value in the revised manuscript .

The term “normalized” in the captions of Figure 4 and Figure 5 is not explicitly explained.

Ans: Thanks for bringing this up. The original manuscript did omit a description of term “normalized”. “normalized” here refers to a statistical strategy applied to the ground truth and predicted RMSF for

analytical purposes, which involves subtracting the mean and dividing by the standard deviation of the original RMSF on PDB atoms. Due to the reorganization of sections, figures 4 and 5 in the original manuscript are now figures 5 and 6 in the revised manuscript. So we add the following description at the end of the captions for Figures 5 and 6:

“Normalization is achieved by subtracting the mean and dividing by the standard deviation of RMSF in the PDB model.”

The term “normalized RMSF” elsewhere retains the same meaning.

Figures 8a, b, c show some redundancy with Figures 8d, e, f.

Ans: Thanks for identifying the issue. There is indeed some redundancy among the subfigures of Figure 8 in the original manuscript. We have redrawn this figure. Due to the rearrangement of sections, it is now Figure 11 in the revised manuscript, as shown below:

p.19 line 2: What does “without the need for a uniform resolution (as a hyperparameter)” specifically refer to?

Ans: Thanks for identifying the issue. We apologize for the unclear expression here. Due to the varying resolutions of cryo-EM maps in the dataset, it is possible to preprocess the maps to a fixed resolution using a filter, as done in DEFMap (Matsumoto *et al.*, 2021). However, if this strategy is adopted, determining the fixed resolution becomes a complex hyperparameter setting issue. The results here demonstrate that these models perform similarly across maps with different resolutions, indicating that our method does not require this preprocessing step. Above all, we have changed this sentence to “without the need to process the maps to a uniform resolution at the preprocessing stage.”, which is in Section 3.4 in the revised manuscript.

p.19 Section 3.6: The last sentence in the first paragraph “Detailed information regarding RMSF-net’s processing time is provided in the Supplementary Materials.” is more appropriate at the end of the next paragraph.

Ans: Thanks for the suggestion. We have moved the sentence to the end of the next paragraph in the revised manuscript, as you suggests.

Supplementary Figure 9: Adding “detailed” or “details” to the first sentence of the caption would be more appropriate.

Ans: Thanks for the suggestion. Due to the addition of contents in the Supplementary Materials, Supplementary Figure 9 in the original manuscript is Supplementary Figure 12 in the revised manuscript. And We have added ‘detailed’ to the first sentence of the caption for Supplementary Figure 12 in the revised manuscript, as follows:

“ Supplementary Figure 12: Detailed relationship of RMSF-net processing time and data sizes.”

p.21 line -3: “For a protein involved in”. This is a general statement, so the plural form of “proteins” should be used.

Ans: Thanks for the suggestion. We have corrected this word as you suggests. This sentence is now in the third paragraph of “Conclusion” section, as follows: “ For proteins involved in physiological

Figure 11: The performance of RMSF prediction methods on different resolution maps in the dataset. (a): Resolution distribution of cryo-EM maps in the dataset (b): Distribution of correlation coefficients of RMSF-net on maps across four resolution ranges. (c): Distribution of correlation coefficients of Occ2RMSF-net across the four resolution ranges. (d): Distribution of correlation difference between RMSF-net and RMSF-net_pdb on the four resolution ranges. The width of the violin in b, c, and d indicates the density of the data, and the central white dot represents the mean of the data. (a): Data distribution of correlation coefficients of RMSF-net on different resolution maps. (b): Data distribution of correlation coefficients of Occ2RMSF-net on different resolution maps. (c): Data distribution of correlation differences between RMSF-net and RMSF-net_pdb on different resolution maps. (d): Distribution of correlation coefficients of RMSF-net on maps across four resolution ranges. (e): Distribution of correlation coefficients of Occ2RMSF-net across the four resolution ranges. (f): Distribution of correlation difference between RMSF-net and RMSF-net_pdb on the four resolution ranges.

activities, RMSF-net enables the assessment of flexibility in distinct regions, identifying areas prone to conformational changes and establishing their functional relevance ”

On both Supplementary Table 1 and Supplementary Table 2, the “preprocessing” time for the first data are much longer than the other data. This may be due to the fact that the program’s warm-up takes up a certain amount of time, and if so, measures should be taken to eliminate that factor.

Ans: Thank you for the comments. Regarding the situation you mentioned, we conducted a careful data check. We concluded that the preprocessing time of the first data significantly exceeds that of the other data, mainly due to its significantly larger data size. Firstly, according to Supplementary Table 1, the total boxes (see the ‘total_box’ column) of the first data are notably higher than those of the

other data. As described in the “Details of RMSF-net processing time” section of the Supplementary Material, there is a linear relationship between total boxes and preprocessing time. After rerunning after program preheating for check, we confirmed that this value is reasonable. To avoid any potential misunderstanding as you mentioned, we have moved this data to the last row in Supplementary Tables 1, 2, and 3. Thanks again for your detailed review.

References

- Abraham, M. J., Murtola, T., Schulz, R., Páll, S., Smith, J. C., Hess, B., and Lindahl, E. (2015). Gromacs: High performance molecular simulations through multi-level parallelism from laptops to supercomputers. *SoftwareX*, **1**, 19–25.
- Baek, M., McHugh, R., Anishchenko, I., Jiang, H., Baker, D., and DiMaio, F. (2024). Accurate prediction of protein–nucleic acid complexes using rosettafoldna. *Nature Methods*, **21**(1), 117–121.
- Chi, X., Jin, X., Chen, Y., Lu, X., Tu, X., Li, X., Zhang, Y., Lei, J., Huang, J., Huang, Z., *et al.* (2020). Structural insights into the gating mechanism of human slc26a9 mediated by its c-terminal sequence. *Cell discovery*, **6**(1), 55.
- Choy, M. S., Li, Y., Machado, L. E., Kunze, M. B., Connors, C. R., Wei, X., Lindorff-Larsen, K., Page, R., and Peti, W. (2017). Conformational rigidity and protein dynamics at distinct timescales regulate ptp1b activity and allostery. *Molecular Cell*, **65**(4), 644–658.
- Dang, S., Van Goor, M. K., Asarnow, D., Wang, Y., Julius, D., Cheng, Y., and van der Wijk, J. (2019). Structural insight into trpv5 channel function and modulation. *Proceedings of the National Academy of Sciences*, **116**(18), 8869–8878.
- Dickson, C. J., Walker, R. C., and Gould, I. R. (2022). Lipid21: complex lipid membrane simulations with amber. *Journal of chemical theory and computation*, **18**(3), 1726–1736.
- Fraser, J. S. and Murcko, M. A. (2024). Structure is beauty, but not always truth. *Cell*, **187**(3), 517–520.
- Gisriel, C. J., Zhou, K., Huang, H.-L., Debus, R. J., Xiong, Y., and Brudvig, G. W. (2020). Cryo-em structure of monomeric photosystem ii from synechocystis sp. pcc 6803 lacking the water-oxidation complex. *Joule*, **4**(10), 2131–2148.
- He, J. and Huang, S.-Y. (2021). Emnuss: a deep learning framework for secondary structure annotation in cryo-em maps. *Briefings in bioinformatics*, **22**(6), bbab156.
- He, J., Lin, P., Chen, J., Cao, H., and Huang, S.-Y. (2022). Model building of protein complexes from intermediate-resolution cryo-em maps with deep learning-guided automatic assembly. *Nature Communications*, **13**(1), 4066.
- Henzler-Wildman, K. and Kern, D. (2007). Dynamic personalities of proteins. *Nature*, **450**(7172), 964–972.
- Jamali, K., Käll, L., Zhang, R., Brown, A., Kimanius, D., and Scheres, S. H. (2024). Automated model building and protein identification in cryo-em maps. *Nature*, pages 1–2.
- Jo, S., Kim, T., Iyer, V. G., and Im, W. (2008). Charmm-gui: a web-based graphical user interface for charmm. *Journal of computational chemistry*, **29**(11), 1859–1865.
- Jones, M., Beuron, F., Borg, A., Nans, A., Earl, C. P., Briggs, D. C., Snijders, A. P., Bowles, M., Morris, E. P., Linch, M., *et al.* (2020). Cryo-em structures of the xpf-ercc1 endonuclease reveal how dna-junction engagement disrupts an auto-inhibited conformation. *Nature communications*, **11**(1), 1120.
- Jumper, J., Evans, R., Pritzel, A., Green, T., Figurnov, M., Ronneberger, O., Tunyasuvunakool, K., Bates, R., Žídek, A., Potapenko, A., *et al.* (2021). Highly accurate protein structure prediction with alphafold. *Nature*, **596**(7873), 583–589.
- Kuhlman, B. and Bradley, P. (2019). Advances in protein structure prediction and design. *Nature reviews molecular cell biology*, **20**(11), 681–697.
- Lane, T. J. (2023). Protein structure prediction has reached the single-structure frontier. *Nature Methods*, **20**(2), 170–173.
- Lawson, C. L., Kryshchovych, A., Adams, P. D., Afonine, P. V., Baker, M. L., Barad, B. A., Bond, P., Burnley, T., Cao, R., Cheng, J., *et al.* (2021). Cryo-em model validation recommendations based on outcomes of the 2019 emdataresource challenge. *Nature methods*, **18**(2), 156–164.
- Li, T., He, J., Cao, H., Zhang, Y., Chen, J., Xiao, Y., and Huang, S.-Y. (2024). All-atom rna structure determination from cryo-em maps. *Nature Biotechnology*, pages 1–9.
- Lomize, M. A., Pogozheva, I. D., Joo, H., Mosberg, H. I., and Lomize, A. L. (2012). Opm database and ppm web server: resources for positioning of proteins in membranes. *Nucleic acids research*, **40**(D1), D370–D376.
- Lugmayr, W., Kotov, V., Goessweiner-Mohr, N., Wald, J., DiMaio, F., and Marlovits, T. C. (2023). Starmap: a user-friendly workflow for rosetta-driven molecular structure refinement. *Nature protocols*, **18**(1), 239–264.
- Maddhuri Venkata Subramaniya, S. R., Terashi, G., and Kihara, D. (2019). Protein secondary structure detection in intermediate-resolution cryo-em maps using deep learning. *Nature Methods*, **16**(9), 911–917.

- Maier, J. A., Martinez, C., Kasavajhala, K., Wickstrom, L., Hauser, K. E., and Simmerling, C. (2015). ff14sb: improving the accuracy of protein side chain and backbone parameters from ff99sb. *Journal of chemical theory and computation*, **11**(8), 3696–3713.
- Marrink, S. J., Corradi, V., Souza, P. C., Ingolfsson, H. I., Tieleman, D. P., and Sansom, M. S. (2019). Computational modeling of realistic cell membranes. *Chemical reviews*, **119**(9), 6184–6226.
- Matsumoto, S., Ishida, S., Araki, M., Kato, T., Terayama, K., and Okuno, Y. (2021). Extraction of protein dynamics information from cryo-em maps using deep learning. *Nature Machine Intelligence*, **3**(2), 153–160.
- Meagher, K. L., Redman, L. T., and Carlson, H. A. (2003). Development of polyphosphate parameters for use with the amber force field. *Journal of computational chemistry*, **24**(9), 1016–1025.
- Morris, K. L., Buffalo, C. Z., Stürzel, C. M., Heusinger, E., Kirchhoff, F., Ren, X., and Hurley, J. H. (2018). Hiv-1 nef is cargo-sensitive ap-1 trimerization switches in tetherin downregulation. *Cell*, **174**(3), 659–671.
- Newport, T. D., Sansom, M. S. P., and Stansfeld, P. J. (2019). The memprotmd database: a resource for membrane-embedded protein structures and their lipid interactions. *Nucleic acids research*, **47**(D1), D390–D397.
- Siebenmorgen, T., Menezes, F., Benassou, S., Merdivan, E., Kesselheim, S., Piraud, M., Theis, F. J., Sattler, M., and Popowicz, G. M. (2023). Misato-machine learning dataset of protein-ligand complexes for structure-based drug discovery. *bioRxiv*, pages 2023–05.
- Terashi, G., Wang, X., Prasad, D., Nakamura, T., and Kihara, D. (2024). Deepmainmast: integrated protocol of protein structure modeling for cryo-em with deep learning and structure prediction. *Nature Methods*, **21**(1), 122–131.
- Van Heel, M., Gowen, B., Matadeen, R., Orlova, E. V., Finn, R., Pape, T., Cohen, D., Stark, H., Schmidt, R., Schatz, M., *et al.* (2000). Single-particle electron cryo-microscopy: towards atomic resolution. *Quarterly reviews of biophysics*, **33**(4), 307–369.
- Vander Meersche, Y., Cretin, G., Gheeraert, A., Gelly, J.-C., and Galochkina, T. (2024). Atlas: protein flexibility description from atomistic molecular dynamics simulations. *Nucleic Acids Research*, **52**(D1), D384–D392.
- Vassal-Stermann, E., Effantin, G., Zubieta, C., Burmeister, W., Iseni, F., Wang, H., Lieber, A., Schoehn, G., and Fender, P. (2019). Cryo-em structure of adenovirus type 3 fibre with desmoglein 2 shows an unusual mode of receptor engagement. *Nature communications*, **10**(1), 1181.
- Walls, A. C., Tortorici, M. A., Bosch, B.-J., Frenz, B., Rottier, P. J., DiMaio, F., Rey, F. A., and Veerles, D. (2016). Cryo-electron microscopy structure of a coronavirus spike glycoprotein trimer. *Nature*, **531**(7592), 114–117.
- Yu, J., Zhang, B., Zhang, Y., Xu, C.-q., Zhuo, W., Ge, J., Li, J., Gao, N., Li, Y., and Yang, M. (2018). A binding-block ion selective mechanism revealed by a na/k selective channel. *Protein & cell*, **9**(7), 629–639.
- Zhang, X., Zhang, B., Freddolino, P. L., and Zhang, Y. (2022). Cr-i-tasser: assemble protein structures from cryo-em density maps using deep convolutional neural networks. *Nature methods*, **19**(2), 195–204.

REVIEWERS' COMMENTS

Reviewer #1 (Remarks to the Author):

The authors have well addressed my concerns and the manuscript has been better improved. I think the revised version of the manuscript is now acceptable. As a significant finding in this paper, I have understood the importance of explicitly giving information of the mean structure of proteins in addition to the raw cryo-EM map, which is a mixture of the mean structure, structural fluctuation, and noise, in predicting RMSF.

Reviewer #1 (Remarks on code availability):

URL described above is in Chinese, and it appears to be unrelated to be RMSF-net. The authors should check it.

Reviewer #2 (Remarks to the Author):

The authors have comprehensively responded and addressed comments. I do not have any comment further.

Reviewer #2 (Remarks on code availability):

Yes, it is fine.

Response to the reviewers

Once again, we sincerely thank the reviewers and editors for their valuable comments and feedback on our work, which greatly improved our work. We have thoroughly reviewed and considered each comment and revised the paper following their comments.

Below, please find the point-by-point response to all the reviewers' comments, where the comments are marked in blue and "Ans:" indicates our answers (in black).

Reviewer 1

Remarks to the Author

The authors have well addressed my concerns and the manuscript has been better improved. I think the revised version of the manuscript is now acceptable. As a significant finding in this paper, I have understood the importance of explicitly giving information of the mean structure of proteins in addition to the raw cryo-EM map, which is a mixture of the mean structure, structural fluctuation, and noise, in predicting RMSF.

Ans: Thank you very much for your insightful comments and suggestions during the review period, which stimulated our thinking and prompted us to further improve the work.

Remarks on code availability

URL described above is in Chinese, and it appears to be unrelated to be RMSF-net. The authors should check it.

Ans: Thank you for the excellent suggestion. We have updated the Code Availability section, set the URL to the GitHub repository of RMSF-net: <https://github.com/XintSong/RMSF-net>.

Reviewer 2

Remarks to the Author

The authors have comprehensively responded and addressed comments. I do not have any comment further.

Ans: Thank you for the positive feedback and constructive comments, which helped us improve the quality of the paper further.

Remarks on code availability

Yes, it is fine.

Ans: Thank you for the feedback.